



# The absorption Angstrom exponent of black carbon with brown coatings: effects of aerosol microphysics and parameterization

Xiaolin Zhang[1,2,3], Mao Mao[1,2], Shihao Tang[3]

[1]Key Laboratory of Meteorological Disaster, Ministry of Education, Nanjing University of Information Science & Technology, Nanjing, 210044, China
[2]School of Atmospheric Physics, Nanjing University of Information Science & Technology, Nanjing, 210044, China
[3]National Satellite Meteorological Center, China Meteorological Administration, Beijing, 100081, China

*Correspondence to*: Mao Mao (mmao@nuist.edu.cn)

**Abstract.** Aerosol absorption Angstrom exponent (AAE) is a crucial optical parameter for their apportionment and characterization. Due to considerable inconsistences associated with observations, a numerical research is a powerful means to give better understanding of the AAE of aged BC aerosols. Numerical studies of the AAE of polydisperse BC aggregates with brown coatings using the exact multiple-sphere T-matrix method (MSTM) are performed. The objective of the study is to thoroughly assess the AAE of coated BC influenced by their observation-based detailed microphysics and then provide a new AAE parameterization for application. At odds with our expectations, BC coated by thin brown carbon with more large particles can have an AAE smaller than 1.0, indicating that BC aerosols internally mixing with brown carbon can even show lower AAE than pure BC particles. The AAE of BC with brown coatings is highly sensitive to absorbing volume fraction of coating, coated volume fraction of BC, shell/core ratio, and particle size distribution with a wide variation, whereas the impacts of BC geometry and BC position within coating are trivial. The AAE of BC with brown coatings can be larger than 3.0, if there are more small coated BC particles, heavy coating, or more brown carbon. However, the AAE of BC with non-absorbing coating shows weakly sensitive to particle microphysics with values around 1.0 (i.e., 0.7–1.4), suggesting the substantial role of absorbing volume fraction of coating in the AAE determination. With more realistic BC geometries, our study also indicates that occurrence of brown carbon may be made confidently unless AAE>1.4. In addition, we present a parameterization of the AAE of coated BC with a size distribution on the basis of numerical results, which can act as a guide for the AAE response to absorbing volume fraction of coating, coated volume fraction of BC, and shell/core ratio. Our findings can improve the understanding and application of the AAE of BC with brown coatings.

## 1 Introduction

Black carbon (BC) aerosols exert a momentous influence on both global and regional climates, on account of their strong scattering and absorption of the radiation [e.g., Guo et al., 2014; Zhang et al., 2015]. Meanwhile, BC tends to be coated by secondary aerosol compositions (such as organics) with the going of aging process, resulting in complicated mixing state and morphology [Schwarz et al., 2008; Myhre, 2009]. Among BC coatings, in addition to non-absorbing components, the



absorbing organics, named brown carbon (BrC), is one type of organic carbon absorbing radiation in the ultraviolet and visible spectra [Clarke et al., 2007]. The largest uncertainty in the estimation of BC climate effects roots from uncertainties in determining their optical properties determined by their complex microphysical properties [Zhang et al., 2019b]. As one of the most significant optical properties, the absorption Angstrom exponent (AAE) describes the wavelength variation in particle absorption, whereas the understanding of aged BC AAE is still limited due to their internally mixing with non-absorptive and absorbing coatings [e.g., Zhang et al., 2008; Shiraiwa et al., 2010]. The lack of accurate understanding and parameterization of the AAE of aged BC has been a pivotal limitation on the assessment of BC radiative effects [e.g., Ramanathan and Carmichael, 2008; Bond et al., 2013].

The concept of AAE is intensively employed for aerosol characterization, as it is considered as a special parameter of aerosol species [e.g., Liu et al., 2018]. For instance, the AAE of pure BC is predicted to be wavelength-independent with a value of 1.0 for particle size less than 50 nm [e.g., Moosmuller et al., 2011], whilst BrC and dust are assumed to have high AAEs [e.g., Russell et al., 2010]. Therefore, in ambient measurements, large AAE is considered to be aerosols originating from dust or biofuel/biomass burning, while small AAE near 1.0 is understood to be BC-rich particles due to the burning of fossil fuel [Russell et al., 2010]. The BrC contributes no absorption at near-infrared wavelengths and absorbs more at shorter wavelengths, whereas BC typically shows an AAE of 1.0 at near-infrared and visible wavelengths [e.g., Kirchstetter and Thatcher, 2012]. Hence, the AAE can be utilized to quantify the separation of BrC absorption from BC absorption based on their distinctive functions of incident wavelength [e.g., Lu et al., 2015]. Additionally, the effects of specified AAE values on the attribution of BC and BrC light absorptions are also investigated [Lack and Langridge, 2013].

For estimating the AAEs of BC particles, numerous experiments have been conducted by measuring their absorption at different wavelengths. The soot from diesel shows an AAE being 1.1, while carbon particles generated from spark exhibit an AAE value of 2.1 [Schnaiter et al., 2003]. After extracting organics, values of AAE ranging from 0.6 to 1.3 for aerosols inside a tunnel or close to a roadway are found [Kirchstetter et al., 2004]. The AAE values of BC-dominated aerosols produced with burning oil, are observed in the range of 0.8–1.1 [e.g., Chakrabarty et al., 2013]. For brown coatings, Kirchstetter et al. [2004] isolate organic carbon with acetone, and an AAE of >4.0 is found for wavelength range of 500–550 nm. Hoffer et al. [2006] extract humic-like substances with water from fine fraction of biomass burning aerosols and introduce an BrC AAE of 6.0–7.0. Clarke et al. [2007] approximate BrC absorption as the difference between total and extrapolated BC absorption with a thermal method to isolate refractory organic carbon, and an BrC AAE as high as 8.0 is estimated. Obviously, reported AAE values of BC particles and their brown coatings are not concordant in disparate studies, and this may be in association with their complicated microphysical properties, such as particle size, mixing state and chemical component.

In spite of providing referential AAE values of BC particles and their brown coatings by the measurements, causes of the AAEs are generally not clear. For instance, what's the principal factor in particle microphysics that leads to complicated AAE: size distribution, mixing state or composition of BC coating? To our knowledge, the measurements have difficulties in reply to these questions, whereas numerical study is a powerful method that can reveal the mechanism behind complicated



BC AAE. To improve our understanding on the AAEs of pure BC and coated BC, numerical investigations have been carried out. On the Basis of the theory of core-shell Mie, Lack and Cappa [2010] investigate the impacts of brown and clear carbon on BC AAE, and show that BC coated in non-absorbing organics can have an AAE as high as 1.6, which complicates attributing measured absorption to BrC within ambient aerosols. Nonetheless, the core-shell Mie structure is in debate [e.g.,

Cappa et al., 2012], as lacy or compact fractal aggregates are widely accepted for BC geometries [e.g., Luo et al., 2018]. With more realistic geometries of BC aggregates, Liu et al. [2018] numerically investigate the AAEs of bare BC and fully coated BC with non-absorptive coating, and highlight the effect of particle size on the AAE. However, the impact of brown coatings on BC AAE is not clear, and meanwhile, no conclusive results for the influences of coating microphysics on BC AAE have been provided. This limits its applications in aerosol-climate models, radiative transfer and remote sensing, due to

the lack of thorough understanding and further parameterization of the AAE of BC with brown coating affected by their microphysics.

Here, numerical studies of the AAE of polydisperse BC particles with brown coatings are systematically carried out according to our current understandings. An accurate multiple-sphere T-matrix method (MSTM) is used for numerical calculation of coated BC absorption properties and further their AAE values. The aim is to evaluate the influences of particle

microphysics, including absorbing volume fraction of coating, coated volume fraction of BC, BC position within coating, BC fractal dimension, shell/core ratio, and size distribution, on the AAEs of BC particles with brown coatings, which hopefully contributes understanding the BC AAEs and their parameterization for application. The performance of the core-shell Mie model for the AAE of coated BC is also evaluated.

## 2 Methodology

### 2.1 models of coated BC

The geometries of freshly emitted BC particles can be typically described by the fractal aggregate [e.g., Sorensen, 2001; Li et al., 2016], mathematically satisfying the statistic scaling rule with the form following

$$N = k_0 \left( \frac{R_g}{a} \right)^{D_f} , \tag{1}$$

$$R_g = \sqrt{\frac{1}{N} \sum_{i=1}^{N} r_i^2} , \tag{2}$$

where $N$, $k_0$ $R_g$, $a$, and $D_f$ represent the monomer number, fractal prefactor, gyration radius, monomer radius, and fractal dimension, respectively. Due to aging process in ambient air, BC can be coated by other species, such as organics, and becomes more compact [e.g., Coz and Leck, 2011; Tritscher et al., 2011]. It is observed that BC particles can externally attached to, partially coated in, or fully encapsulated in coatings [China et al., 2013, 2015]. This study considers BC





aggregate core with a spherical coating, following the coated BC models built by Zhang et al. [2018], and the sketch maps of three typical coated BC structures considered (i.e., externally attached, partially coated and fully coated) are portrayed in Figure 1.

For coated BC, the coated volume fraction of BC ( $F$ ) is a crucial microphysical parameter characterizing its mixing state, and it follows

$$F = \frac{V_{BC\ inside}}{V_{BC}},\qquad(3)$$

where $V_{BC\ inside}$ and $V_{BC}$ are the volume of BC monomers within coating and overall BC volume, respectively. The details of $F$ and the construction of coated BC models are illustrated in Zhang et al. [2018]. The $k_f$ of BC aggregate in this study is assumed as 1.2 [Sorensen, 2001], and BC absorption does not change substantially with $D_f$ varying from 0.9 to 2.1 [Liu and Mishchenko, 2005]. This study considers $N$ value of 200 for accumulation BC, following the methods described in Zhang et al. [2017, 2018], because it is observed that BC particles are mostly in accumulation mode. Two BC $D_f$ of 2.8 and 1.8 on behalf of compact and lacy BC aggregates, respectively, are considered [e.g., Liu et al., 2018]. The BC shell/core ratio $D_p / D_c$ ranging from 1.1 to 2.7 is assumed based on recent observations shown in Liu et al. [2015] and Zhang et al. [2016].

**2.2 Simulation of coated BC absorption**

Given the models of coated BC built, their orientation-averaged absorptions are accurately computed utilizing the powerful multiple-sphere T-matrix method [Mackowski, 2014]. The MSTM is popularly utilized for plentiful numerical researches of BC optical and radiative properties [e.g., Mishchenko et al., 2016]. As it is meaningful to average bulk aerosol properties over certain size distribution for atmospheric applications, a lognormal particle size distribution for an ensemble of coated BC is assumed following

$$\mathrm{n}(r) = \frac{1}{\sqrt{2\pi} r \ln(\sigma_g)} \exp\left[ -\left( \frac{\ln(r) - \ln(r_g)}{\sqrt{2}\ln(\sigma_g)} \right)^2 \right],\qquad(4)$$

where $\sigma_g$ and $r_g$ demote the geometric standard deviation and geometric mean radius, respectively [e.g., Schwarz et al., 2008]. We assume $r_g$ as 0.075 μm [Yu and Luo, 2009] and $\sigma_g$ to be 1.59 [Zhang et al., 2012], as coated BC in accumulation mode is considered. Considering size distribution, coated BC bulk absorption properties are obtained based on the equation following

$$\left\langle C_{abs} \right\rangle = \int_{r_{\min}}^{r_{\max}} C_{abs}(r) n(r) d(r).\qquad(5)$$

The range of radius is set to be 0.05–0.5 μm with an equidistant interval of 0.005 μm for the averaging.



The absorption properties of coated BC particles are investigated at multiple incident wavelengths between 350 nm and 700 nm in steps of 50 nm. The BC refractive index is normally assumed as wavelength independent in near-visible and visible spectral regions [Moosmuller et al., 2009; Luo et al., 2018], and a typical value of $1.85 - 0.71i$ is considered [Bond and Bergstrom, 2006]. For the refractive index of coating of absorbing organics (i.e., brown carbon), it real part is assumed to be

a constant of 1.55 [Chakrabarty et al., 2010], whereas its imaginary part is substantially dependent on incident wavelength over shorter visible and ultraviolet regions [e.g., Moosmuller et al., 2009; Alexander et al., 2008]. The imaginary parts of BrC refractive indices at different wavelengths assumed in this study follow Kirchstetter et al. [2004].

### 2.3 Calculating the absorption Angstrom exponent of coated BC

Given that bulk absorption cross sections at various wavelengths are obtained, the absorption Angstrom exponent of coated

BC, a microphysical parameter describing the wavelength variation in particle absorption, can be calculated. As particle absorption universally decreases exponentially along with the increase of wavelength over near-infrared and visible spectral region [e.g., Lewis et al., 2008], the AAE is defined in forms of

$$\langle C_{abs}(\lambda) \rangle = C_0 \lambda^{-AAE} \quad \text{or}$$

$$\ln\langle C_{abs}(\lambda) \rangle = \ln(C_0) - AAE \ln(\lambda), \tag{6}$$

where $\lambda$, $C_{abs}$ and $C_0$ denote the incident wavelength, bulk aerosol absorption cross section, and a wavelength-independent constant, respectively. The AAE is normally obtained with particle absorption at two wavelengths following

$$AAE = -\frac{\ln(\langle C_{abs}(\lambda_1) \rangle / \langle C_{abs}(\lambda_2) \rangle)}{\ln(\lambda_1 / \lambda_2)}, \tag{7}$$

where $\langle C_{abs}(\lambda_1) \rangle$ and $\langle C_{abs}(\lambda_2) \rangle$ are the bulk aerosol absorption cross sections at wavelengths of $\lambda_1$ and $\lambda_2$, respectively [e.g., Utry et al., 2014]. Nonetheless, the AAE obtained from Eq. (7) is rather sensitive to observational wavelengths selected,

and notable distinct AAE values for different wavelength ranges can be approximated [Moosmuller and Chakrabarty, 2011]. To acquire the most representational AAE, absorption cross sections of coated BC at eight aforementioned wavelengths are applied, and the best AAE is obtained with fitting log-transformed absorptions over the wavelength spectra based on a linear regression (see the second format in Eq. (6)). Figure 2 gives an example for the AAE calculation, and the bulk absorption cross sections as a function of wavelength are illustrated in logarithmic scale. The black squares in the figure are calculated

by the MSTM for BC particles partially coated by BrC (BC $D_f = 2.8$, $D_p / D_c = 1.9$, and $F = 0.5$) with the aforementioned size distribution and refractive indices. The logarithmic absorption depicts an apparent linear variation in the figure, and the red line as a result of the linear fit comes out. Therefore, the line slope, i.e., 2.1, is the value of AAE of BC coated by BrC with relevant microphysical parameters, and the bias induced by chosen absorptions at two wavelengths may be averted. Note that this example shown gives a good linear relationship, being possibly not true for all coated BC cases, while this fit





method still produces the best AAE representation. Since the AAE of coated BC is acquired, systematic studies of the impacts of brown coating on the AAE of BC particles follow.

## 3 The AAE of BC with brown coatings

### 3.1 Effects of coating structures on the AAE of BC coated by BrC

Due to various microphysical parameters, we first study their impacts on the AAE of BC coated by BrC for a fixed particle size distribution. Figure 3 depicts the AAE of BC aggregates coated by BrC calculated with the aforementioned methods for distinct shell/core ratios and coated volume fractions of BC. The absorption properties are averagely calculated over internal-mixed BC-BrC ensembles with the aforementioned size distribution. BC fractal dimensions of 1.8 and 2.8 and coated volume fractions of BC ranging from 0.0 to 1.0 are selected for the investigation. For fully coated BC aggregates (i.e.,

$F = 1.0$), an outmost off-center core-shell and concentric core-shell are considered in Fig. 3, while the popular core-shell Mie model is also studied for the comparison.

As evident in Fig. 3, the AAEs of BC coated by BrC are sensitive to BC fractal dimension, coated volume fraction of BC, and shell/core ratio. The AAE becomes much stronger with shell/core ratio becoming larger, indicating that thinly coated BC has a small AAE whereas heavy coating results in large AAE. BC particles having larger $F$ value exhibit smaller AAE for

the same BC $D_f$ and $D_p / D_c$. Initiating from BC $D_f$ of 2.8, externally attached BC-BrC particles exhibit large AAE variation of 0.8–3.5 with $D_p / D_c$ ranging from 1.1 to 2.7 (see Fig. 3a). Meanwhile, for partially coated BC showing $F = 0.5$, values of AAE vary from 1.3 to 3.1 with $D_p / D_c$ increasing in the range of 1.5–2.7. When BC aggregates are fully coated by BrC, with the augment of $D_p / D_c$ from 1.9 to 2.7, the AAE alters in the range of 1.5–2.6. The coated BC AAE is slightly sensitive to the BC position within brown coating, and its sensitivity becomes stronger as $D_p / D_c$ becomes larger.

For fixed $D_p / D_c$, the AAEs of coated BC with an off-center core-shell structure are slightly larger than those with a concentric core-shell structure with differences within 0.2 (see Fig. 3e). The AAE of coated BC aggregates is also slightly sensitive to BC $D_f$, and the sensitivity shows weaker as $D_p / D_c$ or $F$ become larger. The AAEs of compact BC coated by BrC are generally smaller than those of lacy coated BC, and the differences are less than 0.3. The core-shell Mie model is widely utilized in aerosol-climate models, whereas its applicability on spectral varying absorption properties has not been

evaluated. Compared to the core-shell Mie model, BC aggregates coated by BrC with various coating microphysics have large AAE values, and this indicates that the assumption of the core-shell Mie model could underestimate BC AAE. Moreover, this underestimation becomes stronger, if BC aggregates with brown coating have smaller $F$. For instance, fully coated BC show slightly larger AAEs than those of concentric spherical coated BC with differences less than 0.2, whereas the differences can be as large as 1.0 for externally attached BC-BrC particles.



On the whole, the impacts of BC fractal dimension and BC position within brown coating on the AAE of coated BC are generally negligible. Nevertheless, the AAE of BC coated by BrC is highly sensitive to shell/core ratio and coated volume fraction of BC, and of the two, shell/core ratio plays a more important role in the AAE determination, highlighting the significance of BC shell/core ratio measurement in ambient air. The currently popular core-shell Mie model reasonably approximates the AAE of fully coated BC by BrC, whereas it underestimates the AAE of partially coated or externally attached BC, and underestimates more for smaller coated volume fraction of BC.

### 3.2 Effect of particle size distribution on the AAE of BC coated by BrC

The influence of particle size distribution on the AAE of BC aggregates (BC $D_f = 2.8$) coated by BrC at various shell/core ratios are shown in Figure 4. Coated volume fractions of BC with $F = 0.0$, 0.5, 1.0 corresponding to three typical BC coating states (i.e., externally attached, partially coated and fully coated) are depicted, and BC is considered to be located at the geometric center for fully coated BC. The lognormal size distributions of coated BC with $r_g$ (x axis) in the range of 0.05–0.15 μm and $\sigma_g$ assumed as the aforementioned 1.59 are considered.

As clearly illustrated in Figure 4, the AAE of BC aggregates coated by BrC is rather sensitive to size distribution, besides shell/core ratio and coated volume fraction of BC. The AAE of BC coated by BrC decreases with increasing $r_g$, i.e., coated BC particle being larger. In addition, BC with different coating structures show different dependences on particle size. For the externally attached structure, BC-BrC internal-mixed particles with various distributions give large AAE variation with a range of 0.6–3.6, wherein thinly coated BC generally exhibits small AAE less than 1.0. For BC partially coated by BrC with $F = 0.5$ and fully coated BC with BC at particle geometric center, their AAEs vary in ranges of 0.8–3.3 and 0.9–3.0, respectively. Comparing all three BC coating structures, with $D_p / D_c$ fixed, the AAE of BC coated by BrC shows stronger variations on size distribution when $F$ is larger. Generally, Fig. 4 indicates that BC coated by BrC gives larger AAEs for higher shell/core ratio, less coated volume fraction of BC, or smaller particle size.

### 3.3 Effect of absorbing volume fraction of coating on the AAE of coated BC

The above simulations assume BC coated by BrC, whereas it may be contaminated by non-absorptive organic carbon in ambient air. While the organic coatings in the atmosphere may contain both absorbing and non-absorbing organics, the absorbing volume fraction of coating ($f$) is a vital microphysical parameter characterizing the percentage of BrC in the whole coatings with the equation following:

$$f = \frac{V_{absorbing}}{V_{absorbing} + V_{non-absorbing}}, \tag{8}$$





where $V_{absorbing}$ is the volume of BrC, and $V_{non-absorbing}$ is the volume of non-absorbing coating. The absorbing and non-absorbing organics form internal-mixed coatings with an effective refractive index determined by their volume fractions. The effective refractive index of the internal-mixed coatings is generated on the basis of the popular volume weighted average method, since it provides acceptable absorption properties for coated BC in accumulation mode [Zhang et al., 2019a, 2019b].

For the refractive index of non-absorbing organics, it real part is assumed to be the same as that of absorbing organics (i.e., 1.55), while its imaginary part is considered as 0. The key microphysical parameters of coated BC considered are summarized in Table 1.

Figure 5 shows calculated AAE of coated BC (BC $D_f = 2.8$) with distinct absorbing volume fraction of coating and shell/core ratio. Again, the externally attached BC, partially coated BC with $F = 0.5$, and fully coated BC with BC at

particle geometric center are presented as different BC coating states. The absorption properties of coated BC particles are averaged with aforementioned fixed size distribution (i.e., $r_g = 0.075 \ \mu m$, $\sigma_g = 1.59$). The high sensitivity of coated BC AAE to absorbing volume fraction of coating is clearly depicted, and for fixed $D_p / D_c$ and $F$, the AAE increases exponentially with incremental absorbing volume fraction of coating. If BC coating is non-absorbing organics (i.e., $f = 0.0$), coated BC with various shell/core ratios gives the AAE in ranges of 0.7–1.0, 1.0–1.0, and 1.0–1.2 for $F = 0.0$, 0.5, and 1.0,

respectively. These small coated BC AAEs may potentially explain small AAE observed in the atmosphere [Schnaiter et al., 2005; Chakrabary et al., 2013; Gyawali et al., 2012]. Nevertheless, as $f$ is increased to 1.0, the AAEs for coated BC with $F = 0.0$, 0.5 and 1.0 are enhanced by factors of 3.5, 3.0 and 3.2 for heavy coating with $D_p / D_c = 2.7$, and by factors of 1.3, 1.3 and 1.5 for thin coating, respectively. Furthermore, the AAE of coated BC with a fixed $F$ shows stronger variation on absorbing volume fraction of coating, when shell/core ratio is larger. However, for a fixed $D_p / D_c$, the variation of AAE on

absorbing volume fraction of coating becomes weaker, as $F$ becomes higher.

To reveal the impact of absorbing volume fraction of coating on the AAE of coated BC under different size distributions, Fig. 6 depicts the AAE results of coated BC with the above mentioned three different coating structures. As the AAE of thinly coated BC show small values with a narrow variation, heavy coated BC with a $D_p / D_c$ of 2.7 is considered for this sensitivity study. One can see that, the AAE of coated BC is sensitive to both size distribution and absorbing volume fraction

of coating, and stronger sensitivity of the AAE to size distribution is found when absorbing volume fraction of coating is larger. For non-absorbing coating, the AAE of externally attached BC with $D_p / D_c = 2.7$ is slightly sensitive to size distribution, and its values are between 0.9 and 1.0. With more BC encapsulated in non-absorbing organics, the AAEs tend to be more sensitive to size distribution, and are in ranges of 0.8–1.1 and 0.7–1.4 for $F = 0.5$ and 1.0, respectively. This may be associated with that intensified $F$ enlarges absorption enhancement of BC coated by non-absorbing organics and its AAE

is altered [Zhang et al., 2018]. The AAE of BC coated by non-absorbing organics in our study is coincident with corresponding results presented in Liu et al. [2018]. With incremental absorbing coatings, the AAE increases sharply,





showing large values in ranges of 2.9–3.6, 2.6–3.3 and 1.9–3.0 for BC coated by BrC with $F = 0.0$, 0.5 and 1.0 , respectively.

In general, among all sensitive microphysical parameters of coated BC, the absorbing volume fraction of coating plays a more substantial role in the AAE determination. The AAE of BC with non-absorbing coating shows weakly sensitive to size

distribution, shell/core ratio, and coated volume fraction of BC, and a narrow AAE variation with values around 1.0 (i.e., approximately 0.7–1.4) is seen. However, with increasing absorbing volume fraction of coating, the coated BC AAE increases exponentially and becomes strongly sensitive to size distribution, shell/core ratio, and coated volume fraction of BC with a wide variation. In addition, our results with more realistic geometries indicate that occurrence of BrC can only be made with confidence if the AAE of coated BC is larger than 1.4, as the AAE smaller than 1.4 can not necessarily exclude

BrC as an important contributor to particle absorption. This is generally in consistent with the findings of Lack and Cappa [2010] produced by a core-shell Mie model, showing that BrC cannot be assigned with confidence unless AAE>～1.6.

### 3.4 Parameterization of the AAE of coated BC

After sensitivity analysis of all microphysical factors in previous subsections, it apparently becomes feasible to parameterize the AAE of coated BC on the basis of the decomposition of the impacts of their microphysics. Among all microphysical

parameters of coated BC, the absorbing volume fraction of coating, coated volume fraction of BC, shell/core ratio, and particle size distribution have significant effects on the AAE, whereas the impacts of BC fractal dimension and BC position within coating are comparatively negligible. Compared to absorbing volume fraction of coating, coated volume fraction of BC, and shell/core ratio, the effect of particle size distribution on the AAE is comparatively complicated. Meanwhile, the impact of size distribution on the AAE is generally smaller than other three important microphysical parameters, and shows

close sensitivities to the absorbing volume fraction of coating and shell/core ratio only for fully coated BC. Thus, to make the parameterization doable, the absorbing volume fraction of coating, coated volume fraction of BC, and shell/core ratio are used for the AAE parameterization, whereas the size distribution is considered independently (i.e., to be fixed). As discussed previously, the absorbing volume fraction of coating, coated volume fraction of BC, and shell/core ratio show clearly monotonic impact on the AAE but to varying degrees. With other microphysical parameters fixed, the AAE of coated BC

varies exponentially with each of these three parameters (i.e., $f$ , $D_p / D_c$ , and $F$ ), and can be well fitted by exponential functions. To be more specific, for a fixed size distribution, the AAE of coated BC is assumedly expressed by

$$AAE = AAE_0 + k_1 e^{k_2 f} + k_3 e^{k_4 D_p / D_c} + k_5 e^{k_6 F} , \qquad (9)$$

where k1–k6 denote the coefficients, indicating the significance of relevant influence on the AAE of coated BC. Considering the aforementioned fixed size distribution (i.e., $r_g = 0.075 \, \mu m$ , $\sigma_g = 1.59$ ), which is commonly utilized in aerosol-climate

models, the coefficients can be fitted and the AAE of coated BC is given by

$$AAE = 6.04 - 1.34 e^{-2.51 f} - 6.12 e^{-0.47 D_p / D_c} - 1.08 e^{0.46 F} . \qquad (10)$$





The fitting coefficients in Equation (10) are acquired based on the smallest root-mean-square relative errors for all calculated values of AAE ( $R^2 = 0.86$ ). The correlation coefficient for three variables (i.e., $f$ , $D_p / D_c$ , and $F$ ) is mildly smaller than that for one variable (i.e., each of $f$ , $D_p / D_c$ , and $F$ ), and this is possibly associated with the lack of considering the combined effects of $f$ , $D_p / D_c$ , and $F$ on the AAE in the parameterization. The influences of particle microphysics on the

AAE of coated BC are obviously confirmed by corresponding coefficients in Equation (10). High absolute values of fitting coefficients imply showing more significant impacts on the AAE of coated BC, while their negative or positive sign means the correlation sign. It is evident that the coated BC AAE is more sensitive to $f$ and $D_p / D_c$ than $F$ .

To confirm the capability of the expresses to approximate the AAE of coated BC, Fig. 7 demonstrates absolute differences between the AAEs approximated by Eq. (10) and those from exact numerical simulations. It is clear that most of the

approximated AAE results are in good agreement with the exact simulations with differences less than 0.2. The externally-attached BC shows relatively poor agreements compared to the partially and fully coated BC particles. However, considering that the partly and fully coated morphologies are dominated in aged BC based on observations [China et al., 2013, 2015], Fig. 7 reveals that this simple parameterization method we proposed gives rather accurate estimations of the AAE of coated BC with typical size distributions.

### 3.5 Atmospheric implications

The theoretical results presented may show universal significance, as inorganic species contained in ambient BC coatings, exhibit similar refractive indices to those of non-absorptive organics in this study. It is found that the absorbing volume fraction of coating plays a crucial role in the AAE determination, and the AAE of BC with non-absorbing coating shows

weakly sensitive to particle microphysics with values around 1.0 (i.e., 0.7–1.4). However, the AAE of BC with brown coatings is highly sensitive to absorbing volume fraction of coating, coated volume fraction of BC, shell/core ratio, and particle size distribution with a wide variation, while the effects of BC fractal dimension and BC position within coating are negligible. With BrC contained in coating, the AAE can be larger than 3.0 for coated BC with more small particles, heavy coating, or more BrC. Although the volume of BrC seems to be responsible for the large AAE of coated BC, more BC

encapsulated in brown coating or more large coated BC particles reduce this effect. Interestingly, BC coated by thin BrC with a large size distribution (i.e., large $r_g$ ) can have the AAE smaller than 1.0, and this implies that BC aerosols containing BrC can even show lower AAE than pure BC particles, which challenge conventional beliefs. Meanwhile, the ambient measurement of small AAE (such as values near 1.0) does not exclude important contributions from BrC absorption to absorption, since BC particles coated by BrC can give similar small AAE values based on our exact numerical simulation.

Our results with more realistic geometries also indicate that occurrence of BrC may be made confidently unless AAE>1.4, which is a replenishment of related findings of Lack and Cappa [2010] produced by the core-shell Mie model. Furthermore,





the impacts of aerosol microphysics on coated BC AAE can be understood by Eq. (9), and can be quantitatively obtained by Eq. (10) for coated BC under a typical lognormal size distribution (i.e., $r_g = 0.075\ \mu\text{m}$, and $\sigma_g = 1.59$). The effects of various microphysical parameters on coated BC AAE are rather complex, and finding a best AAE parameterization is difficult. Nevertheless, with the help of Equations (9–10), the AAE of coated BC with brown coatings may be easily

obtained if its key microphysics (i.e., size distribution, shell/core ratio, absorbing volume fraction of coating, and coated volume fraction of BC) are known.

## 4 Conclusions

The study numerically investigates the AAEs of polydisperse BC particles with brown coatings affected by their microphysics that are constrained within realistic ranges based on observations. The BC morphology is modelled by the

fractal aggregate, whereas the MSTM is employed to exactly calculate the light absorptions of coated BC. The results reveal that the AAE of BC with brown coatings is highly sensitive to absorbing volume fraction of coating, coated volume fraction of BC, shell/core ratio, and particle size distribution with a broad variation, while the influences of BC position within coating and BC fractal dimension are generally negligible. Nevertheless, the AAE of BC with non-absorbing coating depicts weakly sensitive to particle microphysics with values around 1.0 (i.e., 0.7–1.4), indicating the substantial role of absorbing

volume fraction of coating in determining the AAE. The AAE of BC with brown coatings can be larger than 3.0 for coated BC with more small particles, heavy coating, or more brown carbon. Meanwhile, BC coated by thin brown carbon with a large size distribution can show an AAE smaller than 1.0, implying that BC aerosols containing brown carbon can even show lower AAE than pure BC particles, and this challenges conventional beliefs. With more realistic geometries, our study also indicates that occurrence of brown carbon may be made confidently unless AAE>1.4, which is a replenishment of

corresponding findings of Lack and Cappa [2010] based on the core-shell Mie model.

Although it is challenging to parameterize the AAE of coated BC based on various microphysical properties, we present a simple AAE parameterization method with key parameters (i.e., coated volume fraction of BC, shell/core ratio, and absorbing volume fraction of coating) for known particle size distribution. For a typical lognormal size distribution (i.e., $r_g = 0.075\ \mu\text{m}$, and $\sigma_g = 1.59$), the complicated influences of the sensitive microphysical parameters on the AAE of coated

BC can be quantified with a good fitting correlation coefficient ($R^2 = 0.86$). Overall, the work clearly demonstrates the distinctive importance of diverse microphysical properties on the AAE of coated BC. Nevertheless, it may need to take caution with our results as a guide, since the ambient observations of microphysical properties of aged BC is still limited at present.




*Author contribution.* M. Mao and X. Zhang conceived the research plan. X. Zhang performed the simulations and wrote the manuscript. All authors discussed the results and contributed the final paper.

*Acknowledgements.* The work is financially supported by the National Natural Science Foundation of China (NSFC) (No. 41505127). We particularly appreciate Daniel W. Mackowski for the source of the codes of MSTM 3.0.

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





**Table 1: Key microphysical properties of coated BC aggregates**

| Parameters | | applied values |
|---|---|---|
| $F$ | | 0.0, 0.25, 0.5, 0.75, 1.0 |
| $D_p/D_c$ | | 1.1, 1.5, 1.9, 2.3, 2.7 |
| $f$ | | 0.0, 0.25, 0.5, 0.75, 1.0 |
| BC $D_f$ | | 1.8, 2.8 |
| Size distribution | $r_g$, μm | 0.075 (0.05–0.15) |
| | $\sigma_g$ | 1.59 |

25





(a)  (b)  (c)

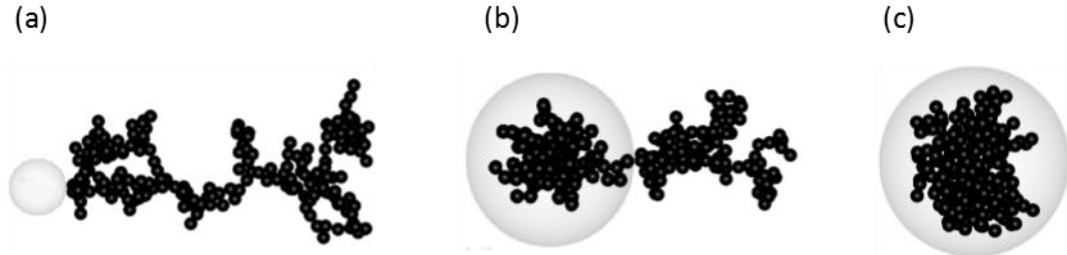

**Figure 1: Sketch maps of geometries of coated black carbon. Examples of fractal black carbon aggregates, containing 200 monomers, are externally attached to (a), partially coated by (b), and fully coated in (c) organics with coated volume fractions of BC being 0.0, 0.5, and 1.0, respectively.**





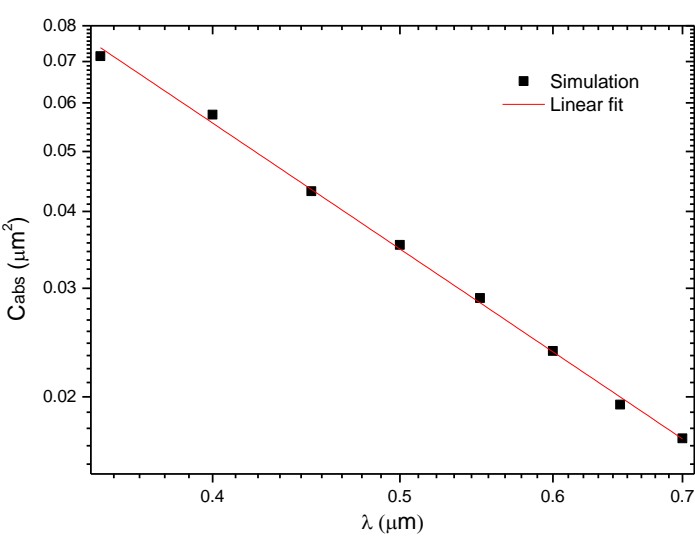

**Figure 2: Absorption cross sections (Cabs) of BC coated by brown carbon as a function of wavelength (λ). The partially coated BC with BC Df of 2.8, F of 0.5, and Dp/Dc of 1.9 is considered as an example.**





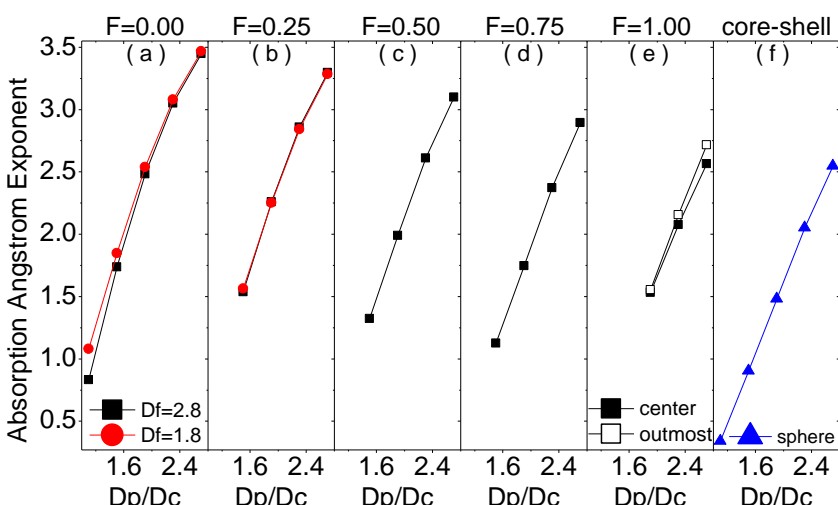

**Figure 3: The absorption Angstrom exponent (AAE) of BC aggregates coated with brown carbon as a function of shell/core ratio (Dp/Dc). The coated volume fractions of BC ($F$) of 0.00, 0.25, 0.5, 0.75, and 1.00, as well as the spherical core-shell structure, are considered (from left to right). Black squares and red circles indicate BC fractal dimensions of ~2.8 and ~1.8, respectively, while blue triangles denote spherical core-shell structures. For coating cases with $F=1$, black solid squares denote BC aggregates located at the particle geometric center, while black open squares indicate BC at an outmost position close to coating boundary.**





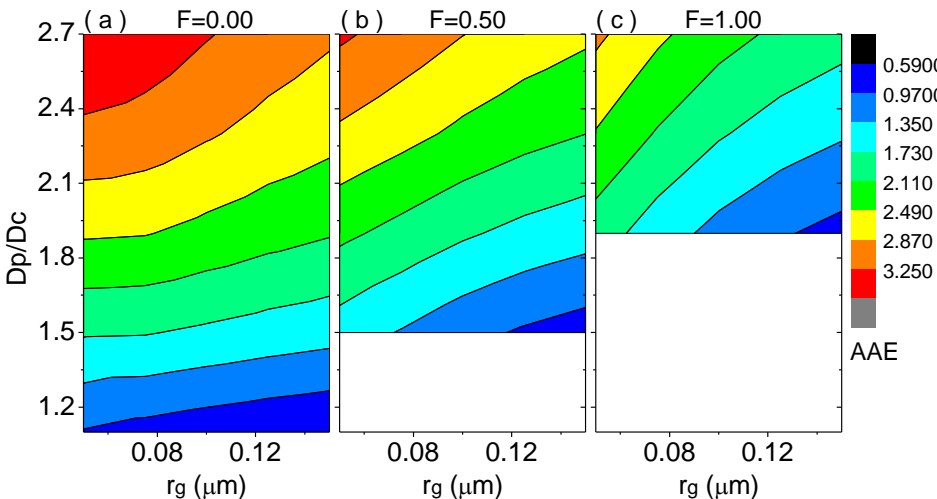

**Figure 4: The absorption Angstrom exponent (AAE) of BC aggregates (BC fractal dimension of ∼2.8) coated by brown carbon with different shell/core ratio (Dp/Dc) and particle size distribution. Three coated volume fractions of BC, being 0.00, 0.50, and 1.00, are shown from left to right. For fully coated BC structure, BC is located at the particle geometric center. The geometric standard deviations (σ$_g$) for used lognormal distribution are 1.59.**



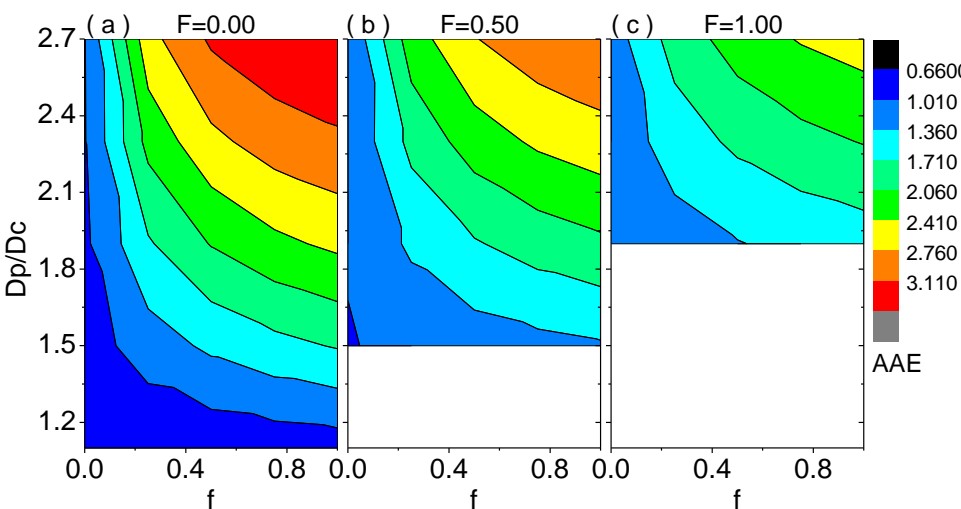

**Figure 5: The absorption Angstrom exponent (AAE) of BC aggregates (BC fractal dimension of ～2.8) coated by organics with different shell/core ratio (Dp/Dc) and absorbing volume fraction of coating (f). Three coated volume fractions of BC, being 0.00, 0.50, and 1.00, are shown from left to right. For fully coated BC structure, BC is located at the particle geometric center.**





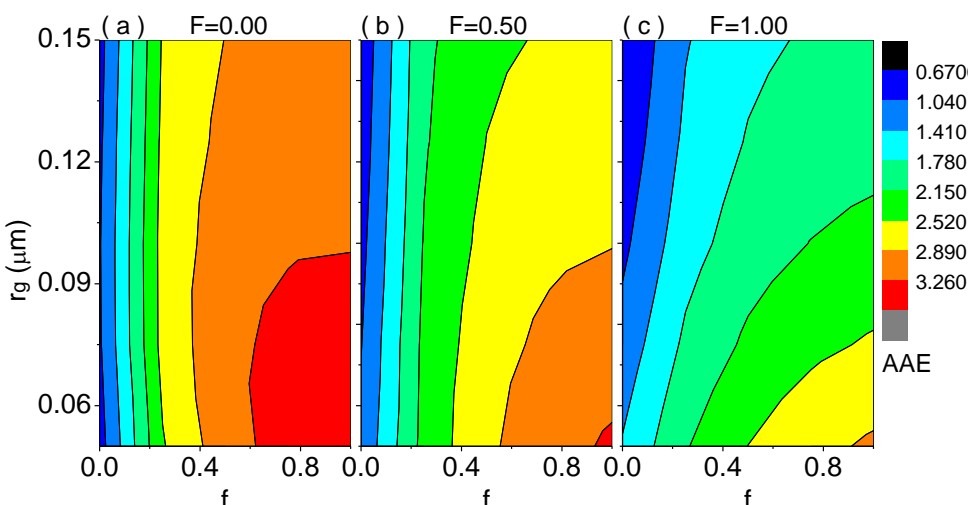

**Figure 6: The absorption Angstrom exponent (AAE) of BC aggregates (BC fractal dimension of ~2.8) coated by organics with different absorbing volume fraction of coating (f) and particle size distribution. Three coated volume fractions of BC, being 0.00, 0.50, and 1.00, are shown from left to right. For fully coated BC structure, BC is located at the particle geometric center. The shell/core ratio is 2.7 and geometric standard deviations ($\sigma_g$) for utilized lognormal distribution are 1.59.**





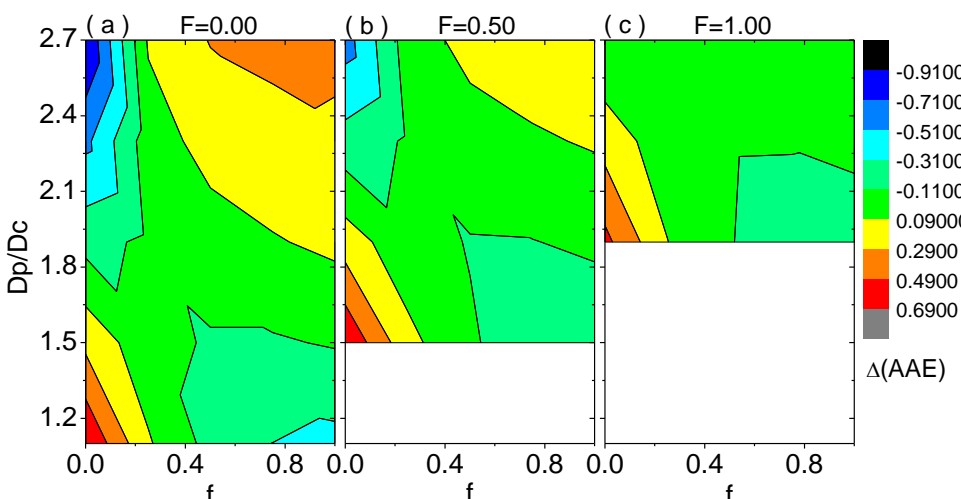

**Figure 7: Absolute differences between the absorption Angstrom exponents (AAEs) approximated by Eq. (10) and those given by accurate numerical simulations. BC aggregates (BC fractal dimension of ～2.8) coated by organics with different shell/core ratio (Dp/Dc) and absorbing volume fraction of coating (f) are considered. Three coated volume fractions of BC, being 0.00, 0.50, and 1.00, are shown from left to right. For fully coated BC structure, BC is located at the particle geometric center.**