# Peer review of "The absorption Angstrom exponent of black carbon with brown coatings: effects of aerosol microphysics and parameterization"

_Atmospheric Chemistry and Physics, 2020_

## Referee Comment (RC1) · Anonymous Referee #1 · 20 Apr 2020

This paper uses the multi-sphere T-matrix method (MSTM) to analyze how BC size, aggregate fractal dimension, and mixing state affects the absorption Angstrom exponent (AAE). The article is well organized and well written, although it could benefit from some minor copy editing in some places. I find it suitable for publication after the corrections listed below.
* * *
MAJOR ISSUES:

INTRODUCTION

[Figure]

The introduction should be expanded somewhat, as there is significant work on this topic that the authors do not mention. For instance, see Liu, JQSRT 2019, Liu and Mishchenko, Rem. Sens. 2018 for aggregated BC computations. I would also search for more. As can be gathered from my comments below, much of the work cited in the intro is not consistent with what I have read in those articles.

Page 2, line 1, authors state: "... the absorbing organics, named brown carbon (BrC), is one type of organic carbon absorbing radiation in the ultraviolet and visible spectra [Clarke et al., 2007]."

This is a little misleading. BrC is not one type of organic carbon; rather, BrC is composed of many different absorbing organic species.

Page 2, line 5, authors state: "The lack of accurate understanding and parameterization of the AAE of aged BC has been a pivotal limitation on the assessment of BC radiative effects [e.g., Ramanathan and Carmichael, 2008; Bond et al., 2013]."

This is very misleading, as these articles do not attribute such large importance to AAE. In fact, I did a search for "Angstrom" in RC08 and did not get a single hit.

P2, L16, authors state: "Hence, the AAE can be utilized to quantify the separation of BrC absorption from BC absorption based on their distinctive functions of incident wavelength [e.g., Lu et al., 2015]."

This is an oversimplification of current AAE discussions, as there are plenty of articles in the literature stating that AAE can not unambigously separate BrC from BC (e.g., see Lack and Cappa (2010) and Lack and Langridge (2013), Schuster ACP 2016, part 2, etc.). If you want readers to take this article seriously, you should highlight the current AAE issues that are being discussed in the literature and then tell readers how your contribution fits into the overall discussion.

P2, L22, authors state: "The AAE values of BC-dominated aerosols produced with burning oil, are observed in the range of 0.8–1.1 [e.g., Chakrabarty et al., 2013]."

But C13 concluded that only mustard oil was dominated by BC, and they measured an average AAE = 1.32 for mustard oil. How did the authors arrive at 0.8-1.1 from the C13 article?

P2, L24: I don't see AAE > 4 anywhere in Kirchstetter 2004.

P2, L26: I don't see BrC AAE ∼8 anywhere in Clarke 2007.

METHOD:

The aggregates used to represent BC in this study seem to have been drawn out of thin air. The authors do not discuss why they chose N=200 or Df = 1.8, 2.8 in detail. Later, the authors draw some fairly broad conclusions based upon this numerical work, but the reader is left wondering how the results might differ if the authors had chosen different aggregates. This is especially important, since the spherical coatings in Figure 1 do not look terribly realistic. How might the results change if the authors used less particles per aggregate (e.g., N = 40, as in Adachi, JGR 2010) and non-spherical coatings? How big are the primary spherules in this work? How would the results change if one alters the spherule sizes? What if one alters N? What role does shielding play? Large N –> more shielding –> less efficient absorption. It would be nice to see one of these aggregate papers address the shielding issue. I realize that shielding is probably to much to add to this paper, but acknowledging that shielding is an important topic that is still unaddressed would be nice.

How do the authors' results compare to other work, such as Liu and Mishchenko (Remote Sensing, 2018)? LM18 computed AAE for particles with many different aggregate configurations and mixing scenarios. Placing the author's results in the context of this wider study could help the reader understand the range of applicability of the results presented here.

The authors frequently state that their calculations are "more realistic," but I have never seen TEM pictures that look like Figure 1b. There are also many articles with non-

spherical aggregate coatings and therefore more realistic than Fig 1c (e.g., Adachi 2010). Many of these articles only address single particles, though.

Also, how do the fractal dimensions Df = 1.8, 2.8 shown in Fig 3 relate to the morphologies shown in Fig 1? That is, what are the Df for the morphologies of Fig 1? More importantly, what do the BC aggregates look like when Df = 1.8, 2.8 and N = 200?

P4, L22: Authors should make clear that these numbers pertain to aggregate sizes, not the monomers. Presumably these radii correspond to equivalent volume spheres, which should also be mentioned. Also, how is r_g related to the gyration radius of Eq 1, R_g?

P5, L28, authors state: "...and the bias induced by chosen absorptions at two wavelengths may be averted."

The authors seem to be stating that the AAE errors are not subject to absolute measurement errors of absorption. However, the AAE is an exponent; as such, it is highly sensitive to absorption measurement errors when AAE is derived from two wavelengths. A simple perturbation analysis using "typical" measurement errors will illustrate this.

RESULTS:

P7, L1, authors state: "On the whole, the impacts of ... BC position within brown coating on the AAE of coated BC are generally negligible."

That's because the shells are not that much larger than the cores (Dp/Dc > 1.6). There are many early papers that investigated the effect of "randomly placed inclusions" vs. a "concentric inclusion." See Fuller JGR 1999, for example. It is worth noting the similarities and differences between your results and the early core/shell work, here.

P7, L23, authors state: "The above simulations assume BC coated by BrC, whereas it may be contaminated by non-absorptive organic carbon in ambient air."
Well, BrC is always "contaminated" with OC. That's because no-one has ever defini-tively separated BrC from OC. For instance, Kirchstetter separated OC from BC, so Kirchstetter's refractive indices represent a mixture of absorbing OC (now widely called BrC) AND non-absorbing OC. These are not two separate compounds, as both BrC and OC represent hundreds (thousands?) of compounds. I believe that this is why there is such a huge range of refractive indices for BrC in the literature. I believe that if anyone ever isolated the absorbing compounds of BrC from other OC, that the resulting BrC refractive index would be higher than the values that the community is using right now.

I really like the concept of this section, but the phrasing is misleading. What you are basically doing is assuming that the Kirchstetter BrC IRI is the upper extreme for BrC absorption, and then considering cases of BrC that are less absorbing than the Kirch-stetter values. You could also look at the range of values provided by other groups as another (perhaps better) way of discussing variable BrC absorption. See Schuster ACP 2016 figures, for instance. Whatever you do, though, the wording should not con-vey the idea that Kirchstetter measured "pure" BrC. I don't believe that K04 meant to convey this.

P9, L8, authors state: "In addition, our results with more realistic geometries indicate that occurrence of BrC can only be made with confidence if the AAE of coated BC is larger than 1.4, as the AAE smaller than 1.4 can not necessarily exclude BrC as an important contributor to particle absorption."

This sentence does not make sense to me.

P10, L25, authors state: "Interestingly, BC coated by thin BrC with a large size dis-tribution (i.e., large $r_g$) can have the AAE smaller than 1.0, and this implies that BC aerosols containing BrC can even show lower AAE than pure BC particles, which challenge conventional beliefs."

Pure and uncoated BC can also have AAE < 1 if the particles are large, according to

Fig 4 when F=0. This corresponds to the geometry of Fig 1a, right? It would be nice if the authors are also able tp present the AAE for a particles that are not touching another sphere, but I believe that they would still obtain AAE < 1 for large aggregates of BC. This should be mentioned here, because AAE is sensitive to particle size. See Fig 6, models 2 & 3 in Liu and Mishchenko (Rem. Sens., 2018); see also Gyawali (ACP, 2009) and Schuster (ACP, 2016).

I don't know what is considered to be "conventional belief," but the AAE = 1 assumption for BC is a by product of the Rayleigh small particle limit for absorption. Aggregates of BC do not necessarily satisfy the "small" criteria, so AAE = 1 does not necessarily hold (especially for collapsed aggregates with significant shielding). Open aggregates can be reasonably modeled as a loose collection of spheres, though, so the AAE = 1 approximations may hold for those cases. Thus, we expect a range of AAE for BC.

Page 10, L30, authors state: "Our results with more realistic geometries also indicate that occurrence of BrC may be made confidently unless AAE>1.4, which is a replenishment of related findings of Lack and Cappa [2010] produced by the core-shell Mie model."

This is exactly opposite of LC2010, per their abstract:

It has often been assumed that observation of an absorption Angstrom exponent (AAE)>1 indicates absorption by a non-BC aerosol. Here, it is shown that BC cores coated in C_Clear can reasonably have an AAE of up to 1.6, a result that complicates the attribution of observed light absorption to C_Brown within ambient particles. However, an AAE<1.6 does not exclude the possibility of C_Brown; rather C_Brown cannot be confidently assigned unless AAE>1.6. – LC2010

CONCLUSIONS:

P11, L16, authors state: "Meanwhile, BC coated by thin brown carbon with a large size distribution can show an AAE smaller than 1.0, implying that BC aerosols containing

brown carbon can even show lower AAE than pure BC particles, and this challenges conventional beliefs."

Here again, a BrC coating is not necessary to achieve AAE < 1.

Also, AAE = 1 for all BC is not a "conventional belief," as many of us know that particle size is important. Lack and Cappa (2010) discuss this, for instance. See also Gyawali (ACP, 2009) and Schuster (ACP, 2016 part 2).

————————————————————-

MINOR ISSUES:

P4, L7, authors state: "...the volume of BC monomers within coating and overall BC volume..." It took me awhile to discern the meaning of this phrase. It would be helpful if the authors point the readers to Fig 1b, here.

P4, L8: k_f has not been defined thus far. Is this the same as the k_0 of Eq 1?

P5, Lines 1-7: This paragraph would be much stronger with an active voice. The authors are discussing things that are "normally" done and providing citations, which sounds like a literature review. The paragraph would be much clearer if the authors tell the reader what they are doing with an active voice; then the citations become the justification.

P5, L10 and throughout: I would avoid using the word "bulk" in this context, as bulk optical properties refer to bulk matter that is much much larger than the wavelength, which is not the topic of this paper.

P2, L9, authors state: "... can be calculated." Here again and throughout – get rid of passive voice. Tell the reader what you did, not what can be done.

P5, line 27: authors state that the slope of the line in Fig 2 is 2.1, but the figure indicates a negative slope. More precise wording is needed.

[Figure]

P6, L12: The authors state that "the AAEs of BC coated by BrC are sensitive to fractal dimension,..." but their Figure 3 indicates that this sensitivity is small when Dp/Dc > 1.5 or so for F =0, and that there is no sensitivity at all when F > 0. This should be mentioned in this paragraph.

P6, L22 and elsewhere: The authors frequently discuss the difference between compact and lacy BC aggregates, but they never tell the reader which Df is more compact (i.e., Df=1.8 or Df=2.8).

Figures 4-7: It is annoying that the colorbar in Figs 4-7 unconventionally decreases upward.

P9, L3 and throughout: "In general, among all sensitive microphysical parameters of coated BC, the absorbing volume fraction of coating plays a more substantial role in the AAE determination."

More substantial than what? Comparative words like 'more' have to be 'more than' something. This seems to happen fairly frequently in this paper (e.g., "more realistic geometries" – more realistic than what?).

P9, Eqs 9 & 10: I don't understand the utility of these empirical equations. The authors are using 3 parameters that are difficult or impossible to measure in order to approximate something that is relatively easy to measure (the AAE). I don't understand the point.

---

## Referee Comment (RC2) · Anonymous Referee #2 · 1 May 2020

The paper describes a numerical study of the Aerosol Absorption Angstrom Exponent (AAE) for aged BC particles. The authors use the multi-sphere T-Matrix method to calculate the optical properties of coated BC particles. One of the "surprising" findings of the study is that, in some circumstances, BC coated by brown carbon exhibits AAE lower than even "pure" BC (I've put quotations because probably there is no such thing as pure BC, apart from a modeling perspective). I think the work is interesting and adds important results useful to the community. Therefore I think the work is worth publishing after the following comments are carefully addressed.

General comments

- The English language should be improved significantly before the manuscript can be published. I would encourage the authors to have a native speaker read over and edit the paper to improve readability. As it is now, grammar and sentence construction issues seriously hamper the readability and therefore the understandability of the paper.

- I found it difficult to clearly understand the different parameters defined in the paper, especially F until much later in the paper. I think it would help a lot to provide the value of F, f, Dp/Dc, Df, etc. and not just the coated volume fractions in Figure 1 and to clearly define these parameters at the very beginning.

- Refractive index: please provide the values used for each wavelength not just references to the literature, maybe provide a table (or a graph) with all the values used (most importantly obviously for BrC.

- It would be interesting to have some sort of physical explanation (or tentative interpretation) for why the Mie calculations result in generally lower AAE.

- The strong dependence of AAE on the shell/core ratio seems quite reasonable because the AAE increases with the increased amount of absorption ascribable to coating, which has a high AAE in the first place, vs. "pure" BC. Less intuitive, but also quite interesting, is maybe the dependence on F.

- For some of the plots, it would be interesting to provide bands instead of point to account of slight variations of different parameters as in a sensitivity study, but I understand that might require a substantial amount of additional work which might not be doable at the time.

- Is there a rationale behind choosing a power laws model vs. a polynomial or any other type of fits for equation 9? I mean, did the authors consider other potential models, or did they pick this one for a specific reason? Also, please provide the fitting parameters' confidence (e.g., 95%) ranges. More on this later (in the specific comments)

- Related to the previous comment, the proposed parametrization does a decent job in the middle of the ranges of f and Dp/Dc, but not so well at all at the extreme values. Although the authors mention that in passing, I think this is an important caveat to point out very clearly in the paper, including in the abstract so that future research will use caution in applying the model for cases it might not be applicable to (for example for F=0, Dp/Dc higher than 2.5 and f near zero, the parametrization-numerical simulation difference in AAE is about 1, which is a very large discrepancy, and 0f 0.5 at the other extreme of Dp/Dc)

Specific comments

Lines 14-16, page 1. The sentence describes an important finding, but I think it is a bit confusing. The reader might ask if the AAE<1 is for BC thinly coated by BC, or BC thickly coated by some other material, or BC coated by a large amount of BrC, or BC coated by a thin layer of BrC and then further coated by a large amount of other material. I would suggest clarifying the sentence.

Line 18, page 1: By "trivial" do the authors mean negligible?

Line 19, page 1: "more small coated BC..." and "more brown carbon..." the comparative "more" should always be accompanied by a clear indication of what we should compare with. In other words, "more" than what or with respect to what? Also "more small" should be "smaller"

Line 20, page 1: "...shows weakly sensitive..." consider rephrasing. Maybe "shows weakly sensitivity..." or "appears to be weakly sensitive..." or similar.

Lines 12-13, page 2: "...AAE is considered to be aerosols originating..." consider revising the wording, this makes it appear as if AAE is an aerosol, while it is the property of the aerosol.

Line 9, page 3: "This limits its applications..." what does "its" refer to?

Lines 6 and 7, page 4: the definition of F is not very clear to me. What does "BC

monomers within coating" mean?

Line 11, page 5: I would not say that "absorption universally decreases exponentially". The power law is a useful practical tool, an approximation, but I would definitely not say that it is a universal law for the wavelength dependence of absorption.

Line 20, page 5: The sentence is not clear.

Line 28, page 5: "the bias induced by chosen absorptions at two wavelengths may be averted". This sentence is not clear. What bias? How is "averted"?

Lines 1 and 2, page 6: I don't understand the sentence "Since the AAE of coated BC is acquired, systematic studies of the impacts of brown coating on the AAE of BC particles follow".

Line 7, page 6: what does "averagely" mean in this context?

Line 18, page 6: "...with the augment of Dp/Dc from 1.9 to 2.7, the AAE alters in the range of 1.5–2" awkward wording, consider revising. What is the "argument of Dp/Dc", what does it mean "AAE alters..."

Lines 9 and 10, page 6: "...an outmost off-center core-shell and concentric core-shell..." is not completely clear to me what the authors refer to. Maybe a drawing similar to Figure 1 or a direct reference to the existing figure 1 (if relevant) would help to understand what exactly is the configuration considered.

Lines 4 to 6, page 7: I think this is an important finding that is worth highlighting (e.g., in the abstract).

Section 3.2, page 7: (a) Does the size distribution refer to the BC component or to the entire mixed particle (BC plus BrC size)? (b) Is the dependence on size distribution evaluated only for the high fractal dimension case? Did the authors also look at the dependence for low fractal dimension? It would be interesting to see the results. (c) Also, did the authors explore potential dependencies on the width of the distribution

(sigma g)?

Lines 9, 10, page 7: The definition of F is provided more clearly here than initially. This definition should be provided much earlier on in the paper.

Line 23, page 7: I would not consider this to be a "contamination"

Lines 25 to the end of page 7: f is finally defined here. I think a reference to its meaning earlier on would help the readability of the paper.

Line 4, page 9: "shows weakly sensitive…" maybe should be "show weak sensitivity" or "is weakly sensitive"

Line 10, page 23" "remove "in" from "This is generally in consistent with the findings…"

Line 21, page 9: I suggest put the defined parameters in parenthesis to assure a clear understanding of what is what even if previously defined already. Such as in: "the absorbing volume fraction of coating (f), coated volume fraction of BC (F), and shell/core ratio (Dp/Dc)"

Line 22, page 9: "…whereas the size distribution is considered independently (i.e., to be fixed)." This is not clear to me.

Line 25, page 9: Maybe "power laws" is more appropriate than "exponential".

Lines 2-4, page 10: This finding and explanation are confusing to me.

Lines 4 to 5, page 10: "The influences of particle microphysics on the AAE of coated BC are obviously confirmed by corresponding coefficients in Equation 5 (10)." I am not sure I understand this sentence. Do the authors mean that the coefficients are large and therefore the dependence is strong, or something else? I guess that becomes clearer in the following sentences.

Line 8, page 10: "…the capability of the expresses…" what does that mean?

Line 12, page 10: "dominated" maybe should be "dominant"? Also, the fully coated

morphologies might be dominant in many circumstances such as biomass burning plumes, but not always, for example not always in urban environments.

Lines 24-25, page 10: "Although the volume of BrC seems to be responsible for the large AAE of coated BC, more BC encapsulated in brown coating or more large coated BC particles reduce this effect." This seems reasonable, what matters more is the volume ratio because that is the determinant variable that splits between the absorption being dominated by BC with low wavelength dependence (low AAE) and the absorption due to the coating (with high AAE for BrC coating). More counter-intuitive, but also interesting seems to be the following sentence; is there any hypothesis on why that might be (meaning why the AAE might be significantly lower than 1 for thin BrC coatings)?

Line 30, page 10: "might be made..." or "might not be made...". Same in the conclusion section.

Line 31, page 10: "which is a replenishment of related findings" consider rewording, the use of "replenishment" here does not seems to be the most appropriate.

Figure 5-7: How does f differ (or how is related to) Dp/Dc?

Figure 7: That is an interesting comparison. It seems like the model does well for intermediate values of f and Dp/Dc values. The model does less well at the extremely lower or higher values of f or Dp/Dc. This might suggest a bias in the model that tends to fit better the center but less well the tails. That might also be due to the power-law fit choice, so, as mentioned in the general comments, it could be good to also explore other parametrizations (such as a polynomial or even just a simple multiple variable linear regression or so) to understand if the power fit is truly justified and appropriate, or if a different model would perform better.

Table 1: Re-define what the different parameters are in the caption so the reader does not have to search for the definitions in the text. F, Dp, Dc, f, etc.

---

## Author Comment (AC1) · 2 Jun 2020

First of all, we would like to thank the anonymous reviewer for his/her thoughtful review and valuable comments to the manuscript. In the revision, we have accommodated all the suggested changes into consideration and revised the manuscript accordingly. All changes are highlighted in RED in the revision. In this point-to-point response, the reviewer's comments are copied as texts in BLACK, and our responses are followed in BLUE.

[Figure]

This paper uses the multi-sphere T-matrix method (MSTM) to analyze how BC size, aggregate fractal dimension, and mixing state affects the absorption Angstrom exponent (AAE). The article is well organized and well written, although it could benefit from some minor copy editing in some places. I find it suitable for publication after the corrections listed below.

Response: Thanks for the constructive comments. The comments are significantly helpful to improve the manuscript, and make the paper more solid. The following presents our point-to-point responses as well as the revision for the manuscript.

The introduction should be expanded somewhat, as there is significant work on this topic that the authors do not mention. For instance, see Liu, JQSRT 2019, Liu and Mishchenko, Rem. Sens. 2018 for aggregated BC computations. I would also search for more. As can be gathered from my comments below, much of the work cited in the intro is not consistent with what I have read in those articles.

Response: For the comments regarding inconsistence of cited articles, we will present in following point-to-point responses. We have added the aggregate BC computations in the Introduction, and cited both papers as follows. "Nonetheless, the core-shell Mie structure is in debate [e.g., Cappa et al., 2012], as lacy or compact fractal aggregates are widely accepted for BC geometries [e.g., Liu and Mishchenko, 2018; Liu et al., 2019]."

Page 2, line 1, authors state: "... the absorbing organics, named brown carbon (BrC), is one type of organic carbon absorbing radiation in the ultraviolet and visible spectra [Clarke et al., 2007]." This is a little misleading. BrC is not one type of organic carbon; rather, BrC is composed of many different absorbing organic species.

Response: We have modified it accordingly in the revision as: "Among BC coatings, in addition to non-absorbing components, the absorbing organics, named brown carbon (BrC), absorbs radiation in the ultraviolet and visible spectra [Clarke et al., 2007]."

Page 2, line 5, authors state: "The lack of accurate understanding and parameterization of the AAE of aged BC has been a pivotal limitation on the assessment of BC radiative effects [e.g., Ramanathan and Carmichael, 2008; Bond et al., 2013]." This is very misleading, as these articles do not attribute such large importance to AAE. In fact, I did a search for "Angstrom" in RC08 and did not get a single hit.

Response: We have revised it accordingly, and abandoned citing both articles as the following: "The lack of accurate understanding and parameterization of the AAE of aged BC has been a pivotal limitation on the assessment of BC radiative effects."

P2, L16, authors state: "Hence, the AAE can be utilized to quantify the separation of BrC absorption from BC absorption based on their distinctive functions of incident wavelength [e.g., Lu et al., 2015]." This is an oversimplification of current AAE discussions, as there are plenty of articles in the literature stating that AAE can not unambigously separate BrC from BC (e.g., see Lack and Cappa (2010) and Lack and Langridge (2013), Schuster ACP 2016, part 2, etc.). If you want readers to take this article seriously, you should highlight the current AAE issues that are being discussed in the literature and then tell readers how your contribution fits into the overall discussion.

Response: Thanks for the constructive comments. We have modified it accordingly following: "The AAE cannot unambiguously be utilized to quantify the separation of BrC absorption from BC absorption despite of their distinctive functions of incident wavelength [e.g., Schuster et al., 2016]."

P2, L22, authors state: "The AAE values of BC-dominated aerosols produced with burning oil, are observed in the range of 0.8–1.1 [e.g., Chakrabarty et al., 2013]." But C13 concluded that only mustard oil was dominated by BC, and they measured an average AAE = 1.32 for mustard oil. How did the authors arrive at 0.8-1.1 from the C13 article?

Response: Thanks for the careful check from the reviewer. We have revised it accordingly as the following: "The AAE of BC-dominated aerosols produced with mustard oil,

is observed to be ï¡đ1.3 [e.g., Chakrabarty et al., 2013]"

P2, L24: I don't see AAE > 4 anywhere in Kirchstetter 2004. P2, L26: I don't see BrC AAE ï¡đ8 anywhere in Clarke 2007.

Response: These AAE values not directly shown in the articles are based on our estimations, and we have abandoned citing both references.

METHOD: The aggregates used to represent BC in this study seem to have been drawn out of thin air. The authors do not discuss why they chose N=200 or Df = 1.8, 2.8 in detail. Later, the authors draw some fairly broad conclusions based upon this numerical work, but the reader is left wondering how the results might differ if the authors had chosen different aggregates. This is especially important, since the spherical coatings in Figure 1 do not look terribly realistic. How might the results change if the authors used less particles per aggregate (e.g., N = 40, as in Adachi, JGR 2010) and non-spherical coatings? How big are the primary spherules in this work? How would the results change if one alters the spherule sizes? What if one alters N? What role does shielding play? Large N –> more shielding –> less efficient absorption. It would be nice to see one of these aggregate papers address the shielding issue. I realize that shielding is probably too much to add to this paper, but acknowledging that shielding is an important topic that is still unaddressed would be nice.

Response: Thanks for the concerns from the reviewer. The aggregates used follow our previous papers, and detailed microphysical parameters and construction of coated BC aggregates have been illustrated therein (such as Zhang et al., 2018, 2019). Meanwhile, used parameters of coated BC aggregates (N=200 or Df = 1.8, 2.8) are commonly seen in various papers, such as Liu et al., 2017; Doner et al., 2017; Teng et al., 2019; Zeng et al., 2019. N=200 is often applied to model BC aggregate at accumulation mode, while BC Df of 2.8 and 1.8 represent compact and lacy BC aggregates, respectively. For the monomer size, we follow Zhang et al., 2018. Only the accumulation mode is considered, as BC is observed to be mostly in accumulation mode. For

the accumulation mode considered, the radius range is set as 0.05–0.5 $\mu$m in steps of 0.005 $\mu$m for the averaging. The exact sizes of BC aggregates can be known based on these coated BC sizes and shell/core ratios. For the effects of BC monomer size or monomer number on the absorption of coated BC aggregates, both are the same question actually as we consider polydisperse coated BC aggregates with lognormal size distributions. For coated BC aggregates with a fixed lognormal size distribution, more BC monomer number corresponds to smaller BC monomer size. We take N=1 as an extreme example (this is core-shell model with a spherical BC core), the absorption of fully coated BC aggregates is almost the same (see Fig. 2a in Zhang et al., 2017). So it is expected that the effects of BC monomer size or monomer number on our AAE results of polydisperse coated BC aggregates are trivial. We assume spherical coatings, but it doesn't mean that the organics is a homogeneous sphere within the overall partially coated BC particle (the organics is a homogeneous sphere only in the case with F=0.0). To build the particle model of partially coated BC, we first generate a BC fractal aggregate and a homogeneous organics sphere, and after BC coated by organics, some BC monomers (volume fraction of F within all BC monomers) will take the place of some organics within the original homogeneous organics sphere. The assumption that the organics are spherical, are based on three aspects in this study. Firstly, the exact numerical method, MSTM, employed in this study is robust and fast in the calculation of optical properties of fractal BC particles, which is in the framework of the T-matrix method. Another powerful DDA method is almost two orders of magnitude slower than the MSTM for coated BC, as shown in Liu et al. [2017]. But the MSTM has the only limitation that the spherical surfaces are nonoverlapping (i.e., for spheres or a cluster of spheres). Secondly, no representative morphology of coating of organics is observed for ambient aged BC aerosols. Some observations of individual aged BC particles actually do show the spherical coating geometry [e.g., Alexander et al., 2008; Zhang et al., 2008; Wu et al., 2016], although some coatings may depict other geometries. While the fractal aggregates have been successfully employed to model BC geometries, simulating the geometry of organics for coated BC is difficult.

Thirdly, however, for coated BC, the simple spherical coating is found to have similar effects on the optical properties to those based on more complicated coating structure [e.g., Dong et al., 2015; Liu et al., 2015; Liu et al., 2017]. Therefore, it is expected that similar absorption results and further AAE will be presented if the BC aggregates are modeled with a non-spherical coating. For the shielding effect, we have mentioned this important topic in the revision as: "The shielding effect of N on the absorption of BC aggregates is an important topic, as larger N can induce more shielding and result in less efficient absorption [Liu and Mishchenko, 2007]." References: Alexander, D. T. L., Crozier, P. A., and Anderson, J. R.: Brown Carbon Spheres in East Asian Outflow and their Optical Properties, Science, 321, 833–836, 2008. Doner, N., Liu, F., and You, J.: Impact of necking and overlapping on radiative properties of coated soot aggregates, Aerosol Sci. Tech., 51, 532–542, 2017. Dong, J., Zhao, J. M., and Liu, L. H.: Morpho-logical effects on the radiative properties of soot aerosols in different internally mixing states with sulfate, J. Quant. Spectrosc. Radiat. Transfer, 165, 43–55, 2015. Liu, C., Li, J., Yin, Y., Zhu, B., and Feng, Q.: Optical properties of black carbon aggregates with non-absorptive coating, J. Quant. Spectrosc. Radiat. Transfer, 187, 443–452, 2017. Liu, F., Yon, J., and Bescond, A.: On the radiative properties of soot aggregates - Part2: effects of coating, J. Quant. Spectrosc. Radiat. Transfer, 172, 134–145, 2015. Teng, S., Liu, C., Schnaiter, M., Chakrabarty, R. K., and Liu, F.: Accounting for the effects of nonideal minor structures on the optical properties of black carbon aerosols, Atmos. Chem. Phys., 19, 2917-2931, 2019. Wu, Y., Cheng, T. H., Zheng, L. J., and Chen, H. Optical properties of the semi-external mixture composed of sulfate particle and different quantities of soot aggregates, J. Quant. Spectrosc. Radiat. Transfer, 179, 139–148, 2016. Zhang, R., Khalizov, A. F., Pagels, J., Zhang, D., Xue, H., and McMurry, P. H.: Variability in morphology, hygroscopicity, and optical properties of soot aerosols during atmospheric processing, P. Natl. Acad. Sci. U.S.A., 105, 10291–10296, 2008. Zhang, X., Mao, M., Yin, Y., and Wang, B.: Absorption enhancement of aged black carbon aerosols affected by their microphysics: A numerical investigation, J. Quant. Spectrosc. Radiat. Transfer, 202, 90–97, 2017. Zhang, X., Mao, M., Yin, Y., and Wang,

B.: Numerical investigation on absorption enhancement of black carbon aerosols partially coated with nonabsorbing organics, J. Geophys. Res., 123, 1297–1308, 2018. Zhang, X., Mao, M., and Yin, Y.: Optically effective complex refractive index of coated black carbon aerosols: from numerical aspects, Atmos. Chem. Phys., 19, 7507–7518, 2019 Zeng, C., Liu, C., Li, J., Zhu, B., Yin, Y., and Wang, Y.: Optical properties and radiative forcing of aged BC due to hygroscopic growth: Effects of aggregate structure, J. Geophys. Res., 124, 4620-4633, 2019.

How do the authors' results compare to other work, such as Liu and Mishchenko (Remote Sensing, 2018)? LM18 computed AAE for particles with many different aggregate configurations and mixing scenarios. Placing the author's results in the context of this wider study could help the reader understand the range of applicability of the results presented here.

Response: We have compared our results with this important work as the following: "The AAE of BC coated by non-absorbing organics in our study is coincident with corresponding results presented in Liu C. et al. [2018] and Liu L. et al. [2018]."

The authors frequently state that their calculations are "more realistic," but I have never seen TEM pictures that look like Figure 1b. There are also many articles with non-spherical aggregate coatings and therefore more realistic than Fig 1c (e.g., Adachi 2010). Many of these articles only address single particles, though. Also, how do the fractal dimensions Df = 1.8, 2.8 shown in Fig 3 relate to the morphologies shown in Fig 1? That is, what are the Df for the morphologies of Fig 1? More importantly, what do the BC aggregates look like when Df = 1.8, 2.8 and N = 200?

Response: Thanks for the constructive comments. Some observations of individual aged BC particles actually do show the spherical coating geometry [e.g., Alexander et al., 2008; Zhang et al., 2008; Wu et al., 2016], which generally look like Figure 1b. Moreover, for coated BC, the simple spherical coating is found to have similar effects on the optical properties to those based on more complicated coating structure [e.g.,

Dong et al., 2015; Liu et al., 2015; Liu et al., 2017]. Therefore, it is expected that similar results of absorption and further AAE will be presented if the BC aggregates are modeled with a non-spherical coating. The BC aggregate shown in Fig. 1a has a Df of 1.8, while its Df is 2.8 in Fig. 1c. The BC aggregate in Fig.1 has N=200, and BC Df of 2.8 and 1.8 represent compact and lacy BC aggregates, respectively. For References, see previous Response.

P4, L22: Authors should make clear that these numbers pertain to aggregate sizes, not the monomers. Presumably these radii correspond to equivalent volume spheres, which should also be mentioned. Also, how is r_g related to the gyration radius of Eq 1, R_g?

Response: We have revised accordingly and mentioned these in the revision as: "Coated BC follows this size distribution, while r is the radius of equivalent volume sphere that has the same volume as that of coated BC aggregate. The exact sizes of BC aggregates can be known on the basis of these coated BC sizes and shell/core ratios." The rg in the size distribution is spherical volume-equivalent radius, which is different from the gyration radius Rg in Equation (2).

P5, L28, authors state: "...and the bias induced by chosen absorptions at two wavelengths may be averted." The authors seem to be stating that the AAE errors are not subject to absolute measurement errors of absorption. However, the AAE is an exponent; as such, it is highly sensitive to absorption measurement errors when AAE is derived from two wavelengths. A simple perturbation analysis using "typical" measurement errors will illustrate this.

Response: We acknowledge that the absolute measurement errors of absorption can induce AAE errors, whereas what we try to talk about here is the issue of wavelength selection. We have modified it to make it clear as the following: "Nonetheless, the AAE obtained from Eq. (7) is rather sensitive to observational wavelengths selected, and notable distinct AAE values can be obtained for different wavelength ranges [Moosmuller

and Chakrabarty, 2011]."

RESULTS: P7, L1, authors state: "On the whole, the impacts of ... BC position within brown coating on the AAE of coated BC are generally negligible." That's because the shells are not that much larger than the cores (Dp/Dc > 1.6). There are many early papers that investigated the effect of "randomly placed inclusions" vs. a "concentric inclusion." See Fuller JGR 1999, for example. It is worth noting the similarities and differences between your results and the early core/shell work, here.

Response: Thanks for the suggestion from the reviewer, whereas we are sorry that we cannot find this old paper (Fuller JGR 1999?) for a comparison.

P7, L23, authors state: "The above simulations assume BC coated by BrC, whereas it may be contaminated by non-absorptive organic carbon in ambient air." Well, BrC is always "contaminated" with OC. That's because no-one has ever definitively separated BrC from OC. For instance, Kirchstetter separated OC from BC, so Kirchstetter's refractive indices represent a mixture of absorbing OC (now widely called BrC) AND non-absorbing OC. These are not two separate compounds, as both BrC and OC represent hundreds (thousands?) of compounds. I believe that this is why there is such a huge range of refractive indices for BrC in the literature. I believe that if anyone ever isolated the absorbing compounds of BrC from other OC, that the resulting BrC refractive index would be higher than the values that the community is using right now. I really like the concept of this section, but the phrasing is misleading. What you are basically doing is assuming that the Kirchstetter BrC IRI is the upper extreme for BrC absorption, and then considering cases of BrC that are less absorbing than the Kirchstetter values. You could also look at the range of values provided by other groups as another (perhaps better) way of discussing variable BrC absorption. See Schuster ACP 2016 figures, for instance. Whatever you do, though, the wording should not convey the idea that Kirchstetter measured "pure" BrC. I don't believe that K04 meant to convey this.

Response: Thanks for the constructive comments from the reviewer, and we have

revised accordingly following: "It should be noticed that no one has ever definitively separated BrC from organic carbon, and to a certain extent, the concept of f here may be treated as that the cases of BrC with imaginary parts of refractive indices less than those of Kirchstetter et al. [2004] are considered due to a range of BrC refractive indices being provided [Schuster et al., 2016]."

P9, L8, authors state: "In addition, our results with more realistic geometries indicate that occurrence of BrC can only be made with confidence if the AAE of coated BC is larger than 1.4, as the AAE smaller than 1.4 can not necessarily exclude BrC as an important contributor to particle absorption." This sentence does not make sense to me.

Response: We have revised it to make it clear as the following: "In addition, our results with more realistic geometries indicate that occurrence of BrC can only be made with confidence if the AAE of coated BC is larger than 1.4."

P10, L25, authors state: "Interestingly, BC coated by thin BrC with a large size distribution (i.e., large $r\_g$ ) can have the AAE smaller than 1.0, and this implies that BC aerosols containing BrC can even show lower AAE than pure BC particles, which challenge conventional beliefs." Pure and uncoated BC can also have AAE < 1 if the particles are large, according to Fig 4 when F=0. This corresponds to the geometry of Fig 1a, right? It would be nice if the authors are also able to present the AAE for a particle that are not touching another sphere, but I believe that they would still obtain AAE < 1 for large aggregates of BC. This should be mentioned here, because AAE is sensitive to particle size. See Fig 6, models 2 & 3 in Liu and Mishchenko (Rem. Sens., 2018); see also Gyawali (ACP, 2009) and Schuster (ACP, 2016). I don't know what is considered to be "conventional belief," but the AAE = 1 assumption for BC is a by product of the Rayleigh small particle limit for absorption. Aggregates of BC do not necessarily satisfy the "small" criteria, so AAE = 1 does not necessarily hold (especially for collapsed aggregates with significant shielding). Open aggregates can be reasonably modeled as a loose collection of spheres, though, so the AAE = 1 approximations

may hold for those cases. Thus, we expect a range of AAE for BC.

Response: Thanks for the reviewer's constructive comments. The results here not only correspond to the geometry of Fig. 1a (i.e., F = 0.0), but also relate to other geometries (see Fig. 4 and Fig. 6). The conventional belief here is that BC aerosols containing BrC should show larger AAE than pure BC particles.

Page 10, L30, authors state: "Our results with more realistic geometries also indicate that occurrence of BrC may be made confidently unless AAE>1.4, which is a replenishment of related findings of Lack and Cappa [2010] produced by the core-shell Mie model." This is exactly opposite of LC2010, per their abstract: It has often been assumed that observation of an absorption Angstrom exponent (AAE)>1 indicates absorption by a non-BC aerosol. Here, it is shown that BC cores coated in C_Clear can reasonably have an AAE of up to 1.6, a result that complicates the attribution of observed light absorption to C_Brown within ambient particles. However, an AAE<1.6 does not exclude the possibility of C_Brown; rather C_Brown cannot be confidently assigned unless AAE>1.6. – LC2010

Response: We have revised it accordingly and abandoned the comparison in this way following: "Our results with more realistic geometries also indicate that occurrence of BrC may not be made confidently unless AAE>1.4."

CONCLUSIONS: P11, L16, authors state: "Meanwhile, BC coated by thin brown carbon with a large size distribution can show an AAE smaller than 1.0, implying that BC aerosols containing brown carbon can even show lower AAE than pure BC particles, and this challenges conventional beliefs." Here again, a BrC coating is not necessary to achieve AAE < 1. Also, AAE = 1 for all BC is not a "conventional belief," as many of us know that particle size is important. Lack and Cappa (2010) discuss this, for instance. See also Gyawali (ACP, 2009) and Schuster (ACP, 2016 part 2).

Response: Thanks for the constructive comments. The results here not only correspond to the geometry of Fig. 1a (i.e., F = 0.0), but also relate to other geometries (see

Fig. 4 and Fig. 6). The conventional belief here is that BC aerosols containing BrC should show larger AAE than pure BC particles.

MINOR ISSUES: P4, L7, authors state: "...the volume of BC monomers within coating and overall BC volume..." It took me awhile to discern the meaning of this phrase. It would be helpful if the authors point the readers to Fig 1b, here.

Response: We have modified it accordingly in the revision as: "where VBC inside and VBC are the volume of BC monomers encapsulated in coating and overall BC volume, respectively (see Fig. 1)."

P4, L8: k_f has not been defined thus far. Is this the same as the k_0 of Eq 1?

Response: We have changed kf to k0.

P5, Lines 1-7: This paragraph would be much stronger with an active voice. The authors are discussing things that are "normally" done and providing citations, which sounds like a literature review. The paragraph would be much clearer if the authors tell the reader what they are doing with an active voice; then the citations become the justification.

Response: We have revised it accordingly following: "We investigate absorption properties of coated BC particles at multiple incident wavelengths between 350 nm and 700 nm in steps of 50 nm. We consider a typical BC refractive index of 1.85-0.71i [Bond and Bergstrom, 2006], as it is normally assumed as wavelength independent in near-visible and visible spectral regions [Moosmuller et al., 2009; Luo et al., 2018]. For the refractive index of coating of absorbing organics (i.e., brown carbon), this study assumes its real part to be a constant of 1.55 [Chakrabarty et al., 2010], whereas its imaginary part is substantially dependent on incident wavelength over shorter visible and ultraviolet regions [e.g., Moosmuller et al., 2009; Alexander et al., 2008]. The imaginary parts of BrC refractive indices at different wavelengths assumed in this study follow Kirchstetter et al. [2004], and are shown in Fig. S1."

[Figure]

P5, L10 and throughout: I would avoid using the word "bulk" in this context, as bulk optical properties refer to bulk matter that is much much larger than the wavelength, which is not the topic of this paper.

Response: We have deleted "bulk" in the revision accordingly.

P5, L9, authors state: "... can be calculated." Here again and throughout – get rid of passive voice. Tell the reader what you did, not what can be done.

Response: We have modified it accordingly as: "Given that bulk absorption cross sections at various wavelengths are obtained, we calculate the absorption Angstrom exponent of coated BC, a microphysical parameter describing the wavelength variation in particle absorption."

P5, line 27: authors state that the slope of the line in Fig 2 is 2.1, but the figure indicates a negative slope. More precise wording is needed.

Response: We have modified it accordingly following: "the negative of the line slope".

P6, L12: The authors state that "the AAEs of BC coated by BrC are sensitive to fractal dimension,..." but their Figure 3 indicates that this sensitivity is small when Dp/Dc >1.5 or so for F =0, and that there is no sensitivity at all when F > 0. This should be mentioned in this paragraph.

Response: We have revised it as the following: "The AAE of coated BC aggregates is also slightly sensitive to BC Df, and the sensitivity shows weaker as Dp/Dc or F become larger. The AAEs of compact BC coated by BrC (i.e., BC Df=2.8) are generally smaller than those of lacy coated BC (i.e., BC Df=1.8) with differences less than 0.3, and there is almost no sensitivity of AAE to BC Df for F>0."

P6, L22 and elsewhere: The authors frequently discuss the difference between compact and lacy BC aggregates, but they never tell the reader which Df is more compact (i.e., Df=1.8 or Df=2.8).
Response: We have revised it accordingly following: "The AAE of coated BC aggregates is also slightly sensitive to BC Df, and the sensitivity shows weaker as Dp/Dc or F become larger. The AAEs of compact BC coated by BrC (i.e., BC Df=2.8) are generally smaller than those of lacy coated BC (i.e., BC Df=1.8) with differences less than 0.3, and there is almost no sensitivity of AAE to BC Df for F>0."

Figures 4-7: It is annoying that the colorbar in Figs 4-7 unconventionally decreases upward.

Response: We consider coated BC microphysical parameters with many discrete points in the numerical study as shown in Table 1, and this may be the reason why the color bars do not look perfectly smooth.

P9, L3 and throughout: "In general, among all sensitive microphysical parameters of coated BC, the absorbing volume fraction of coating plays a more substantial role in the AAE determination." More substantial than what? Comparative words like 'more' have to be 'more than' something. This seems to happen fairly frequently in this paper (e.g., "more realistic geometries" – more realistic than what?).

Response: Thanks for the comments, and we have modified it accordingly as: "In general, the absorbing volume fraction of coating plays a more substantial role in the AAE determination than other sensitive microphysical parameters."

P9, Eqs 9 & 10: I don't understand the utility of these empirical equations. The authors are using 3 parameters that are difficult or impossible to measure in order to approximate something that is relatively easy to measure (the AAE). I don't understand the point.

Response: Thanks for the concerns from the reviewer. There are considerable inconsistences associated with AAE observations, and the uncertainties in absorption measurements at multi-wavelengths (such as using aethalometer) may be one significant reason. The Equations 9 and 10 can be act as the AAE response to the key sensitive

microphysical parameters (i.e., absorbing volume fraction of coating, coated volume fraction of BC, and shell/core ratio) for a quantitative understanding. Moreover, the absorbing volume fraction of coating may be acquired with the chemical measurements by a single particle aerosol mass spectrometry (SPAMS) (e.g., Wang et al., 2019). The coated volume fraction of BC can be observed with a scanning electron microscopy (SEM) (e.g., China et al., 2013, 2015), while the shell/core ratio can be obtained using a single-particle soot photometer (SP2) (e.g., Liu et al., 2015; Zhang et al., 2016).

Anonymous Referee #2

First of all, we would like to thank the anonymous reviewer for his/her thoughtful review and valuable comments to the manuscript. In the revision, we have accommodated all the suggested changes into consideration and revised the manuscript accordingly. All changes are highlighted in RED in the revision. In this point-to-point response, the reviewer's comments are copied as texts in BLACK, and our responses are followed in BLUE.

The paper describes a numerical study of the Aerosol Absorption Angstrom Exponent (AAE) for aged BC particles. The authors use the multi-sphere T-Matrix method to calculate the optical properties of coated BC particles. One of the "surprising" findings of the study is that, in some circumstances, BC coated by brown carbon exhibits AAE lower than even "pure" BC (I've put quotations because probably there is no such thing as pure BC, apart from a modeling perspective). I think the work is interesting and adds important results useful to the community. Therefore I think the work is worth publishing after the following comments are carefully addressed.

Response: Thanks for the constructive comments. The comments are significantly helpful to improve the manuscript, and make the paper more solid. The following presents our point-to-point responses as well as the revision for the manuscript.

General comments - The English language should be improved significantly before the manuscript can be published. I would encourage the authors to have a native

speaker read over and edit the paper to improve readability. As it is now, grammar and sentence construction issues seriously hamper the readability and therefore the understandability of the paper.

Response: Thanks for the constructive comment, and the manuscript has been polished by an English editor.

- I found it difficult to clearly understand the different parameters defined in the paper, especially F until much later in the paper. I think it would help a lot to provide the value of F, f, Dp/Dc, Df, etc. and not just the coated volume fractions in Figure 1 and to clearly define these parameters at the very beginning.

Response: We have modified accordingly and defined these parameters at the beginning of the Methodology as: "It is observed that BC particles can externally attached to, partially coated in, or fully encapsulated in coatings [China et al., 2013, 2015]. This study considers BC aggregate core with a spherical coating, following the coated BC models built by Zhang et al. [2018], and the sketch maps of three typical coated BC structures considered (i.e., externally attached, partially coated and fully coated) are portrayed in Figure 1. For coated BC, the coated volume fraction of BC (F) is a crucial microphysical parameter characterizing its mixing state, and it follows F=VBC inside/VBC (3) where VBC inside and VBC are the volume of BC monomers encapsulated in coating and overall BC volume, respectively (see Fig. 1). With the definition, the externally attached, partially coated, and fully coated BC aggregates show F=0, 0<F<1, and F=1, respectively. For coated BC, shell/core ratio Dp/Dc is an important microphysical parameter and is defined as spherical equivalent particle diameter divided by BC core diameter (Dc). The fractal dimension Df is a parameter describing the compactness of BC aggregates, and due to aging process in ambient air, BC can be coated by other species, such as organics, becoming compact (i.e., large Df) [e.g., Coz and Leck, 2011; Tritscher et al., 2011]."

- Refractive index: please provide the values used for each wavelength not just references to the literature, maybe provide a table (or a graph) with all the values used (most importantly obviously for BrC.

Response: We have provided a graph for BrC refractive index accordingly in Fig. S1.

- It would be interesting to have some sort of physical explanation (or tentative interpretation) for why the Mie calculations result in generally lower AAE.

Response: Thanks for the constructive suggestion, and we have tried to give some tentative interpretations in the revision as: "This is probably due to that the absorption of BC coated by BrC with core-shell Mie model show slower increase with decreased wavelength than that of coated BC with realistic particle geometry."

- The strong dependence of AAE on the shell/core ratio seems quite reasonable because the AAE increases with the increased amount of absorption ascribable to coating, which has a high AAE in the first place, vs. "pure" BC. Less intuitive, but also quite interesting, is maybe the dependence on F.

Response: Thanks for the reviewer's constructive comments.

- For some of the plots, it would be interesting to provide bands instead of point to account of slight variations of different parameters as in a sensitivity study, but I understand that might require a substantial amount of additional work which might not be doable at the time.

Response: Thanks for the constructive comments, and we will do it in our further work in the future as it requires a substantial amount of additional work which might not be doable at the time.

- Is there a rationale behind choosing a power laws model vs. a polynomial or any other type of fits for equation 9? I mean, did the authors consider other potential models, or did they pick this one for a specific reason? Also, please provide the fitting parameters' confidence (e.g., 95%) ranges. More on this later (in the specific comments)

Response: Thanks for the constructive comments from the reviewer. The parameterization of the AAE of coated BC is challenging and difficult, as there are three microphysical parameters (i.e., F, Dp/Dc and f) for fitting (Common fitting is only for one parameter). The rationale behind choosing a fit type may be that we first have to select a model to well fit one parameter and then guess the possible model for fitting three parameters. We have tried the polynomial fit and multiple variable linear regression fit, and both are failed (The polynomial fit even cannot converge, while linear regression fit show very low correlation coefficient R2). Therefore, the power law model shown already is the best one at present. For the fitting parameters' confidence ranges, we have added accordingly in the revision as: "the coefficients (with 95% confidence range) can be fitted and the AAE of coated BC is given by".

- Related to the previous comment, the proposed parametrization does a decent job in the middle of the ranges of f and Dp/Dc, but not so well at all at the extreme values. Although the authors mention that in passing, I think this is an important caveat to point out very clearly in the paper, including in the abstract so that future research will use caution in applying the model for cases it might not be applicable to (for example for F=0, Dp/Dc higher than 2.5 and f near zero, the parametrization-numerical simulation difference in AAE is about 1, which is a very large discrepancy, and 0f 0.5 at the other extreme of Dp/Dc)

Response: Thanks for the constructive comments. We have pointed it out in the Abstract as: "The proposed parameterization of coated BC AAE does a decent prediction for moderate BC microphysics, whereas caution should be taken in applying it for extreme cases, such as externally attached coated BC morphology."

Specific comments Lines 14-16, page 1. The sentence describes an important finding, but I think it is a bit confusing. The reader might ask if the AAE<1 is for BC thinly coated by BC, or BC thickly coated by some other material, or BC coated by a large amount of BrC, or BC coated by a thin layer of BrC and then further coated by a large amount of other material. I would suggest clarifying the sentence.

Response: We have modified it accordingly as: "more large BC particles coated by thin brown carbon can have an AAE smaller than 1.0".

Line 18, page 1: By "trivial" do the authors mean negligible?

Response: We have modified "trivial" to "negligible".

Line 19, page 1: "more small coated BC: : :" and "more brown carbon: : :" the comparative "more" should always be accompanied by a clear indication of what we should compare with. In other words, "more" than what or with respect to what? Also "more small" should be "smaller"

Response: We have revised it accordingly as: "if there are plenty of small coated BC particles, heavy coating, or a large amount of brown carbon".

Line 20, page 1: ": : :shows weakly sensitive: : :" consider rephrasing. Maybe "shows weakly sensitivity: : :" or "appears to be weakly sensitive: : :" or similar.

Response: We have revised "shows weakly sensitive" to "appears to be weakly sensitive".

Lines 12-13, page 2: ": : :AAE is considered to be aerosols originating: : :" consider revising the wording, this makes it appear as if AAE is an aerosol, while it is the property of the aerosol.

Response: We have revised it in the revision as: "Therefore, in ambient measurements, large AAE is considered to be that aerosols originate from dust or biofuel/biomass burning, while small AAE near 1.0 is understood to be that aerosols are BC-rich particles due to the burning of fossil fuel [Russell et al., 2010]."

Line 9, page 3: "This limits its applications: : :" what does "its" refer to?

Response: We have revised "its" to "the AAE".

Lines 6 and 7, page 4: the definition of F is not very clear to me. What does "BC

monomers within coating" mean?

Response: We have changed "BC monomers within coating" to "BC monomers encapsulated in coating" and added "see Fig. 1" to make it clear.

Line 11, page 5: I would not say that "absorption universally decreases exponentially". The power law is a useful practical tool, an approximation, but I would definitely not say that it is a universal law for the wavelength dependence of absorption.

Response: We have deleted "universally".

Line 20, page 5: The sentence is not clear.

Response: We have modified it in the revision as: "Nonetheless, the AAE obtained from Eq. (7) is rather sensitive to observational wavelengths selected, and notable distinct AAE values can be obtained for different wavelength ranges [Moosmuller and Chakrabarty, 2011]."

Line 28, page 5: "the bias induced by chosen absorptions at two wavelengths may be averted". This sentence is not clear. What bias? How is "averted"?

Response: We have modified it accordingly following: "the AAE bias induced by wavelength selection may be averted by this fitting method".

Lines 1 and 2, page 6: I don't understand the sentence "Since the AAE of coated BC is acquired, systematic studies of the impacts of brown coating on the AAE of BC particles follow".

Response: We have deleted this sentence as it is only for a smooth transition.

Line 7, page 6: what does "averagely" mean in this context?

Response: We have deleted "averagely".

Line 18, page 6: ": : :with the augment of Dp/Dc from 1.9 to 2.7, the AAE alters in the range of 1.5–2" awkward wording, consider revising. What is the "argument of Dp/Dc",

what does it mean "AAE alters: : :"

Response: We have revised it following: "When BC aggregates are fully coated by BrC, with the increase of Dp/Dc from 1.9 to 2.7, the AAE varies in a range of 1.5–2.6."

Lines 9 and 10, page 6: ": : :an outmost off-center core-shell and concentric coreshell : : :" is not completely clear to me what the authors refer to. Maybe a drawing similar to Figure 1 or a direct reference to the existing figure 1 (if relevant) would help to understand what exactly is the configuration considered.

Response: We have added a direct reference (i.e., [Zhang et al., 2019]) to help to understand what exactly is the configuration considered.

Lines 4 to 6, page 7: I think this is an important finding that is worth highlighting (e.g., in the abstract).

Response: We have highlighted it in the Abstract "The currently popular core-shell Mie model reasonably approximates the AAE of fully coated BC by brown carbon, whereas it underestimates the AAE of partially coated or externally attached BC, and underestimates more for smaller coated volume fraction of BC."

Section 3.2, page 7: (a) Does the size distribution refer to the BC component or to the entire mixed particle (BC plus BrC size)? (b) Is the dependence on size distribution evaluated only for the high fractal dimension case? Did the authors also look at the dependence for low fractal dimension? It would be interesting to see the results. (c) Also, did the authors explore potential dependencies on the width of the distribution (sigma g)?

Response: Thanks for the reviewer's constructive comments. (a) The size distribution refers to the entire mixed particle, and we have modified it to make it clear as: "The lognormal size distributions for coated BC with rg (x axis) in the range of 0.05–0.15 $\mu$m and $\sigma$g assumed as the aforementioned 1.59 are considered." (b) We have added the AAE dependence on size distribution for low BC fractal dimension, and the comparisons of the AAE between low and high BC fractal dimension are shown in Fig. S2. The differences of the AAE of BC with brown coating induce by BC fractal dimension are generally trivial. (c) We have explored the AAE dependence on the width of particle size distribution, which is shown in Fig. S3. As aerosol-climate models generally consider particle size distributions with fixed width (i.e., $\sigma$g) but varying radius (i.e., rg), we show the AAE dependence on $\sigma$g in Fig. S3. The AAE of BC with brown coating generally decrease with increased width of size distribution, except for externally attached BC-BrC with small width of size distribution (i.e., F=0.0 and $\sigma$g <1.39).

Lines 9, 10, page 7: The definition of F is provided more clearly here than initially. This definition should be provided much earlier on in the paper.

Response: We have provided it earlier in the revision as: "With the definition, the externally attached, partially coated, and fully coated BC aggregates show F=0, 0<F<1, and F=1, respectively.".

Line 23, page 7: I would not consider this to be a "contamination"

Response: We have revised it accordingly as: "The above simulations assume BC coated by BrC, whereas non-absorptive organic carbon can also exist in BC coatings in ambient air."

Lines 25 to the end of page 7: f is finally defined here. I think a reference to its meaning earlier on would help the readability of the paper.

Response: We thank for the reviewer's constructive comment, whereas the absorbing volume fraction of coating (f) is defined by us. It may be the first time (to our knowledge) to define f here, and we are sorry that there is no reference for defining it earlier in the manuscript.

Line 4, page 9: "shows weakly sensitive: : :" maybe should be "show weak sensitivity" or "is weakly sensitive"

Response: We have modified "shows weakly sensitive" to "shows weak sensitivity"

accordingly.

Line 10, page 9" "remove "in" from "This is generally in consistent with the findings: : :"

Response: We have deleted "in" accordingly.

Line 21, page 9: I suggest put the defined parameters in parenthesis to assure a clear understanding of what is what even if previously defined already. Such as in: "the absorbing volume fraction of coating (f), coated volume fraction of BC (F), and shell/core ratio (Dp/Dc)"

Response: We have revised it accordingly following: "Thus, to make the parameterization doable, the absorbing volume fraction of coating (f), coated volume fraction of BC (F), and shell/core ratio (Dp/Dc) are used for the AAE parameterization".

Line 22, page 9: ": : :whereas the size distribution is considered independently (i.e., to be fixed)." This is not clear to me.

Response: We have revised it to make it clear as: "whereas the size distribution is not considered (i.e., to be fixed)".

Line 25, page 9: Maybe "power laws" is more appropriate than "exponential".

Response: We have revised "exponential" to "power law" accordingly.

Lines 2-4, page 10: This finding and explanation are confusing to me.

Response: We have revised it to make it clear in the revision following: "The correlation coefficient for parameterizing with three variables (i.e., f, Dp/Dc, and F) is mildly smaller than that with one variable (i.e., each of f, Dp/Dc, and F), and this is possibly associated with the lack of considering the combined interaction effects of f, Dp/Dc, and F on the AAE in the parameterization."

Lines 4 to 5, page 10: "The influences of particle microphysics on the AAE of coated BC are obviously confirmed by corresponding coefficients in Equation 5 (10)." I am not

sure I understand this sentence. Do the authors mean that the coefficients are large and therefore the dependence is strong, or something else? I guess that becomes clearer in the following sentences.

Response: It means that the coefficients are large and therefore the dependence is strong. Meanwhile, The Equation can be treated as a quantitative understanding of the influences of particle microphysics on the AAE of coated BC. We have revised it to make it clear in the revision as: "The influences of particle microphysics on the AAE of coated BC are obviously confirmed by corresponding coefficients in Equation (10) with a quantitative understanding."

Line 8, page 10: ": : :the capability of the expresses: : :" what does that mean?

Response: We have revised it to make it clear in the revision as: "To confirm the capability of this parameterization in approximating the AAE of coated BC".

Line 12, page 10: "dominated" maybe should be "dominant"? Also, the fully coated morphologies might be dominant in many circumstances such as biomass burning plumes, but not always, for example not always in urban environments.

Response: We have revised it accordingly in the revision as: "considering that the partly and fully coated morphologies are dominant in aged BC based on observations".

Lines 24-25, page 10: "Although the volume of BrC seems to be responsible for the large AAE of coated BC, more BC encapsulated in brown coating or more large coated BC particles reduce this effect." This seems reasonable, what matters more is the volume ratio because that is the determinant variable that splits between the absorption being dominated by BC with low wavelength dependence (low AAE) and the absorption due to the coating (with high AAE for BrC coating). More counter-intuitive, but also interesting seems to be the following sentence; is there any hypothesis on why that might be (meaning why the AAE might be significantly lower than 1 for thin BrC coatings)?

Response: Thanks for the constructive comments from the reviewer. The hypothesis on why that might be is the one we do not know at present and still needs further studies.

Line 30, page 10: "might be made: : :" or "might not be made: : :". Same in the conclusion section.

Response: We have modified "might be made" to "might not be made" accordingly in the revision.

Line 31, page 10: "which is a replenishment of related findings" consider rewording, the use of "replenishment" here does not seems to be the most appropriate.

Response: We have deleted it as we have no appropriate rewording.

Figure 5-7: How does f differ (or how is related to) Dp/Dc?

Response: The f and Dp/Dc are two different microphysical parameters of coated BC, and they show no relations. The f is absorbing volume fraction of coating, characterizing the percentage of BrC in the whole coatings, while Dp/Dc is shell/core ratio of coated BC that is the spherical equivalent particle diameter divided by BC core diameter.

Figure 7: That is an interesting comparison. It seems like the model does well for intermediate values of f and Dp/Dc values. The model does less well at the extremely lower or higher values of f or Dp/Dc. This might suggest a bias in the model that tends to fit better the center but less well the tails. That might also be due to the power-law fit choice, so, as mentioned in the general comments, it could be good to also explore other parametrizations (such as a polynomial or even just a simple multiple variable linear regression or so) to understand if the power fit is truly justified and appropriate, or if a different model would perform better.

Response: Thanks for the constructive comments. The parameterization of the AAE of coated BC is challenging and difficult, because there are three microphysical parameters (i.e., F, Dp/Dc and f) for fitting, and common fitting is only for one parameter. We have tried the polynomial fit and multiple variable linear regression fit, and both are failed (The polynomial fit even cannot converge, while linear regression fit show very low correlation coefficient R2). The power law model shown already is the best one at present.

Table 1: Re-define what the different parameters are in the caption so the reader does not have to search for the definitions in the text. F, Dp, Dc, f, etc.

Response: We have redefined these parameters in Table 1.

Please also note the supplement to this comment: https://www.atmos-chem-phys-discuss.net/acp-2020-224/acp-2020-224-AC1-supplement.pdf

---

## Author Response (AR2)

I think the paper presents highly valuable results and the findings are definitely worth publication. The author did a reasonable job addressing the reviewers' comments and the paper is more readable now. I still think the paper would benefit from some editorial work to improve readability and correct grammar issues; in particular, in the newly added sentences.

**Response:** Thanks for the reviewer's constructive comments. The comments are significantly helpful to improve readability of our manuscript, and make the paper more solid. The following presents our point-to-point responses as well as the revision for the manuscript.

Just a few examples (but many more are present):
1. In the abstract: "more large BC particle..." do the authors mean "BC particles with larger coating" or "a larger number of BC coated particles" or something else? From the paper it seems quite plausible that they mean "BC particles with larger coating", but as it is the sentence can be confusing for the reader as they start reading through the paper. A very similar issue with the sentence "if there are plenty of small coated BC particles...", and "...for smaller coated volume fraction of BC"; does "smaller" refer to the fraction of BC or for BC particles that are smaller in size? Again the interpretation becomes evident after reading the rest of the paper but more clarity since the beginning would help readability.

**Response:** We have modified it accordingly (Page 1, lines 16, 21 and 27).

2. "...the absorbing organics, named brown carbon (BrC), absorbs radiation" maybe should be "the absorbing organics... absorb"?

**Response:** We have modified it accordingly (Page 2, line 6).

3. "Therefore, in ambient measurements, large AAE is considered to be that aerosols originate from dust or biofuel/biomass burning, while small AAE near 1.0 is understood to be that aerosols are BC-rich particles due to the burning of fossil fuel", maybe "Therefore, in ambient measurements, large AAE is considered to indicate that aerosols originate from dust or biofuel/biomass burning, while small AAE near 1.0 is understood to indicate that aerosols are BC-rich particles due to the burning of fossil fuel"

**Response:** We have modified it accordingly (Page 2, lines 16-17).

4. "It is observed that BC particles can externally attached to, partially coated in, or fully encapsulated in coatings [China et al., 2013, 2015]. This study considers BC aggregate core with a spherical coating, following the coated BC models built by Zhang et al. [2018], and the sketch maps of three typical coated BC structures considered (i.e., externally attached, partially coated and fully coated) are portrayed in Figure 1." maybe reword as "It is observed that BC particles can be externally attached to, partially coated in, or fully encapsulated by coatings [China et al., 2013, 2015]. This study considers a BC aggregate core with a spherical coating, following the coated BC models built by Zhang et al. [2018]. The sketch maps of three typical coated BC structures considered here (i.e., externally attached, partially coated, and fully coated) are portrayed in Figure 1."

**Response:** We have modified it accordingly (Page 3, lines 23-25).

5. "The AAE of coated BC aggregates is also slightly sensitive to BC Df, and the sensitivity shows weaker as Dp/Dc or F become larger." maybe could be rephrased as: "The AAE of coated BC aggregates is also slightly sensitive to the Df of BC, and the sensitivity becomes weaker as Dp/Dc or F increase."

**Response:** We have modified it accordingly (Page 7, line 3).

6. "This is probably due to that the absorption of BC coated by BrC with core-shell Mie model show slower increase with decreased wavelength than that of coated BC with realistic particle geometry." maybe could be rephrased as: "This is probably because the absorption of BC coated by BrC calculated from the core-shell Mie model shows a slower increase with decreased wavelength than that calculated using realistic particle geometry."

**Response:** We have modified it accordingly (Page 7, lines 8-10).

7. The sentence "coating. It should be noticed that no one has ever definitively separated BrC from organic carbon, and to a certain extent, the concept of f here may be treated as that the cases of BrC with imaginary parts of refractive indices less than those of Kirchstetter et al. [2004] are considered due to a range of BrC refractive indices being provided [Schuster et al., 2016]." is very confusing, please consider rewording it and maybe breaking it down into two sentences.

**Response:** We have broken it down into two sentences (Page 8, lines 10-13).

8. Line 27, page 8: "small coated BC" again does this indicate small BC particles that are coated or BC particles that have a thin (small) coating?

**Response:** Here "small" indicates small AAE, and we have modified it to make it clear (Page 8, line 27).

9. "The correlation coefficient for parameterizing with three variables (i.e., f , / p c DD, and F ) is mildly smaller than that with one variable (i.e., each of f , / p c DD, and F ), and". I am confused by this sentence, I would guess that the more parameters one uses the higher the correlation coefficient should be, so "is mildly smaller" seems weird... but maybe I am misunderstanding the sentence here. I would consider revising this sentence to make it clearer.

**Response:** We have modified it accordingly (Page 10, line 17).

10. Line 10 page 11: "more large coated BC" see previous comments on these kinds of expressions.

**Response:** We have modified it accordingly (Page 11, line 10).